# Dynamic multifactor hubs interact transiently with sites of active transcription in *Drosophila* embryos

**Mustafa Mir[1†], Michael R Stadler[1,2†], Stephan A Ortiz[1], Colleen E Hannon[1], Melissa M Harrison[3], Xavier Darzacq[1]\*, Michael B Eisen[1,2,4]\***

[1]Department of Molecular and Cell Biology, University of California, Berkeley, Berkeley, United States; [2]Howard Hughes Medical Institute, University of California, Berkeley, Berkeley, United States; [3]Department of Biomolecular Chemistry, University of Wisconsin–Madison, Madison, United States; [4]Department of Integrative Biology, University of California, Berkeley, Berkeley, United States

**Abstract** The regulation of transcription requires the coordination of numerous activities on DNA, yet how transcription factors mediate these activities remains poorly understood. Here, we use lattice light-sheet microscopy to integrate single-molecule and high-speed 4D imaging in developing *Drosophila* embryos to study the nuclear organization and interactions of the key transcription factors Zelda and Bicoid. In contrast to previous studies suggesting stable, cooperative binding, we show that both factors interact with DNA with surprisingly high off-rates. We find that both factors form dynamic subnuclear hubs, and that Bicoid binding is enriched within Zelda hubs. Remarkably, these hubs are both short lived and interact only transiently with sites of active Bicoid-dependent transcription. Based on our observations, we hypothesize that, beyond simply forming bridges between DNA and the transcription machinery, transcription factors can organize other proteins into hubs that transiently drive multiple activities at their gene targets.
**Editorial note:** This article has been through an editorial process in which the authors decide how to respond to the issues raised during peer review. The Reviewing Editor's assessment is that all the issues have been addressed (see decision letter).
DOI: https://doi.org/10.7554/eLife.40497.001

**\*For correspondence:**
darzacq@berkeley.edu (XD);
mbeisen@berkeley.edu (MBE)

[†]These authors contributed equally to this work

**Competing interests:** The authors declare that no competing interests exist.

## Introduction

The earliest stages of animal development are dominated by DNA replication and cell or nuclear division, and are primarily driven by maternally deposited RNAs and proteins. Later, control is transferred to the embryonic genome in the maternal-to-zygotic transition (MZT), during which transcription of the embryonic genome commences while maternal products are degraded (*Harrison and Eisen, 2015*).

The MZT in *Drosophila melanogaster* begins in the early syncytial blastoderm after nine rounds of nuclear division (nuclear cycle 9, nc9) (*Foe and Alberts, 1983*). The number of transcribed genes increases gradually as interphase periods steadily lengthen between cycles 9 and 13, before giving way to full-scale zygotic genome activation (ZGA) coincident with cellularization during the 1 hr long interphase of the 14th nuclear cycle (*Edgar et al., 1986*; *Edgar and Schubiger, 1986*; *Pritchard and Schubiger, 1996*; *Anderson and Lengyel, 1981*; *Zalokar, 1976*).

Thousands of genes become transcriptionally active during the MZT, including several hundred transcribed in defined spatial and temporal patterns along the anterior-posterior (AP) and dorsal-ventral (DV) axes (*Combs and Eisen, 2013*; *Combs and Eisen, 2013*; *Lécuyer et al., 2007*; *Wilk et al., 2016*; *Tomancak et al., 2007*), which serve as the first markers of the nascent body plan

of the developing embryo. The formation of these patterns is directed through interactions of DNA-binding proteins known as transcription factors with non-coding regulatory genomic regions known as enhancers. Enhancers are typically bound by combinations of activating and repressing transcription factors and drive transcription of target genes in patterns that depend on the differential combination of factors present in nuclei at different positions within the embryo. However, beyond this basic paradigm, it remains poorly understood how the composition and arrangement of transcription factor binding at enhancers dictates the output of the genes they regulate and what role interactions among binding factors play in this process.

In recent years, it has become clear that patterning transcription factors are only part of the complex systems that specify enhancer activity. Among the key additional players is the ubiquitously distributed maternal factor Zelda (*Staudt et al., 2006*; *ten Bosch et al., 2006*; *Liang et al., 2008*; *De Renzis et al., 2007*) (Zld, also known as Vielfaltig, Vfl) that we and others have shown plays a central role in the spatio-temporal coordination of gene activation, and in facilitating the binding of patterning factors to their target enhancers (*Harrison et al., 2011*; *Li et al., 2014*; *Nien et al., 2011*; *Foo et al., 2014*; *Schulz et al., 2015*; *Sun et al., 2015*).

Zelda is often described as a 'pioneer' transcription factor (*Zaret and Mango, 2016*), in that its primary function appears to be to facilitate the binding of other factors indirectly by influencing chromatin state. However, how it accomplishes this remains unclear. Zelda has a cluster of four C2H2 Zn-fingers (ZFs) near the C-terminus that mediate its DNA binding activity and two additional ZFs near the N-terminus which have been implicated in controlling its activation potential (*Hamm et al., 2017*), but most of the rest of the protein consists of varying types of low-complexity sequences.

Such low-complexity domains (LCDs) are thought to facilitate protein-protein interactions that mediate the formation of higher order structures, including phase separated domains (*Kato and McKnight, 2018*; *Brangwynne et al., 2009*). There is increasing evidence that higher order structures mediated by LCDs play an important role in transcriptional regulation (*Chong et al., 2018*; *Boehning et al., 2018*; *Strom et al., 2017*; *Kato and McKnight, 2018*), although the precise nature of this role remains less than clear. One hypothesis is that domains formed by homo- and heterotypic interactions between LCDs serve to locally enrich transcription factors, potentially in the vicinity of their targets (*Tsai et al., 2017*), thereby altering their local concentration and modulating their binding dynamics.

We recently explored this idea by utilizing lattice light-sheet microscopy (LLSM) (*Chen et al., 2014a*; *Chen et al., 2014b*) to carry out single-molecule imaging and tracking of eGFP-labeled Bicoid (Bcd)—the primary anterior morphogen in *D. melanogaster*—in living embryos (*Mir et al., 2017*). Bicoid proteins are distributed in a concentration gradient along the anterior-posterior axis, and activate approximately 100 genes in a concentration-dependent manner, primarily in anterior portions of the embryo (*Xu et al., 2014*). The sharpness of the response of Bcd targets to its gradient has led to the proposal of various models of cooperative regulation (*Frohnhöfer and Nüsslein-Volhard, 1986*; *Driever and Nüsslein-Volhard, 1988*), but the molecular basis for this apparent cooperation remains incompletely worked out.

We previously showed that Bcd binds DNA transiently (has a high $k_{off}$) and that its binding is concentrated in discrete sub-nuclear domains of locally high Bcd density that we refer to as 'hubs' (*Mir et al., 2017*). These hubs are more prominent in posterior nuclei where Bcd concentration is low, but in which it still binds specifically to target loci. We proposed that Bcd hubs facilitate binding, especially at low concentrations, by increasing the local concentrations of Bcd in the presence of target loci, thereby increasing $k_{on}$ and factor occupancy (*Mir et al., 2017*).

Prompted by previous observations (*Hannon et al., 2017*) that Bcd can bind to inaccessible chromatin on its own at high concentrations in the anterior but requires Zld to do so at low concentrations in the posterior, we examined the distribution of Bcd binding in nuclei lacking Zld and found that Bcd hubs no longer form (*Mir et al., 2017*). Our preliminary experiments with fluorescently tagged Zld revealed that it also forms hubs (distinct clusters of Zld were also recently reported by *Dufourt et al., 2018*).

The combined observations that Bcd forms hubs that depend on the presence of Zld, and that Zld also forms hubs, motivated us to quantify the spatial and temporal relationships between Zld and Bcd molecules in *Drosophila* embryos. However, these experiments required several advancements in our technical capabilities to both tag and image single molecules.

Here we first describe Cas9-mediated tagging of endogenous loci with bright, photoswitchable fluorescent proteins that provide greatly improved signal-to-noise and tracking abilities, modifications to the LLSM necessary to activate these tagged proteins, and the development of biological and analytical tools to study the interactions between proteins, and also between proteins and sites of active transcription. We use this technological platform to characterize the single-molecule and bulk behavior of Zld and Bcd in isolation, in relation to each other, and to the transcriptional activation of the canonical Bcd target gene, *hunchback* (*hb*).

We find that both Bcd and Zld bind DNA highly transiently, with residence times on the order of seconds. Furthermore, both proteins form high-concentration hubs in interphase nuclei which are highly dynamic and variable in nature. By simultaneously imaging the bulk spatial distribution of Zld (to track hubs) and single molecules of Bcd, we show that Bcd binding is both enriched and stabilized within Zld hubs, an effect that becomes more pronounced at low Bcd concentrations in the embryo posterior. Finally, we explore the functional role of Zld and Bcd hubs in activating *hb* and find that hubs of both proteins interact transiently with the active *hb* locus, with preferential interactions of Bcd hubs with active loci leading to a time-averaged enrichment of the protein at the locus. Collectively our data suggest a model in which dynamic multi-factor hubs regulate transcription through stochastic encounters with target genes.

## Results

### Single-molecule tracking of proteins endogenously tagged with photoactivatable fluorescent proteins

We used Cas9-mediated homologous replacement (*Bassett et al., 2013*; *Bassett et al., 2014*; *Gratz et al., 2013*; *Gratz et al., 2014*; *Yu et al., 2013*; *Baena-Lopez et al., 2013*; *Sebo et al., 2013*; *Kondo and Ueda, 2013*; *Ren et al., 2013*) (*Figure 1—figure supplement 1*) to tag endogenous loci of Bcd and Zld at their N-termini with the photoactivatable fluorescent protein mEos3.2 (*Zhang et al., 2012*), which has high-quantum efficiency, is highly monomeric, and photostable compared to other photoactivatable proteins. Zld was also independently tagged with the bright green fluorescent protein mNeonGreen (*Hostettler et al., 2017*). All tagged lines presented here are homozygous viable and have been maintained as homozygous lines for many generations. As Bicoid and Zelda are both required for viability (Bicoid maternally, Zelda maternally and zygotically), this provides strong support for the functionality of the fluorescently tagged fusion proteins. To account for the possibility that un-tagged protein (arising from internal initiation, cryptic splicing, or proteolytic cleavage) could be responsible for phenotypic viability of insertions, we performed western blots on embryos from tagged lines and verified that detectable amounts of un-tagged protein are not present in any of the the lines (*Figure 1—figure supplement 1*). To serve as a control for single-molecule experiments, we also generated lines containing ubiquitously expressed mEos3.2-tagged Histone H2B (His2B).

To utilize the photoactivatable mEos3.2 for single-molecule tracking, we modified a lattice light-sheet microscope (*Chen et al., 2014a*; *Chen et al., 2014b*) to allow continuous and tunable photo-activation from a 405 nm laser (*Figure 1—figure supplement 2*; *Figure 1—video 1*). We optimized this setup using mEos3.2-Zld, controlling particle density (*Figure 1—figure supplement 3*) to facilitate tracking (*Hansen et al., 2018*; *Izeddin et al., 2014*) and found that we could obtain excellent signal-to-noise ratios sufficient for robust single-molecule detection (*Figure 1A*, *Figure 1—video 2*, *Figure 1—video 3* , *Figure 1—video 4*) and tracking of both mobile and immobile molecules at frame intervals ranging from 10 to 500 ms (*Figure 1B* and *Figure 1—video 5*, *Figure 1—video 6*).

We deployed this platform to perform single-molecule imaging and tracking of Zld, Bcd, and His2B at 10, 100 ms, and 500 ms frame intervals (*Figure 2*, *Figure 2—video 1*, *Figure 2—video 2*, *Figure 2—video 3*). These different temporal resolutions each capture distinct aspects of molecular behavior: short exposure times are sufficient to detect single molecules and fast enough to track even rapidly diffusing molecules (*Figure 2—video 1*). However, because imaging single molecules at high-temporal resolution (10 ms) requires high-excitation illumination, most bound molecules photobleach before they unbind, encumbering the accurate measurement of long binding times. At longer exposure times of 100 ms and 500 ms, fast diffusing proteins are blurred into the background, and lower excitation powers lowers photobleaching rates such that unbinding events can be detected

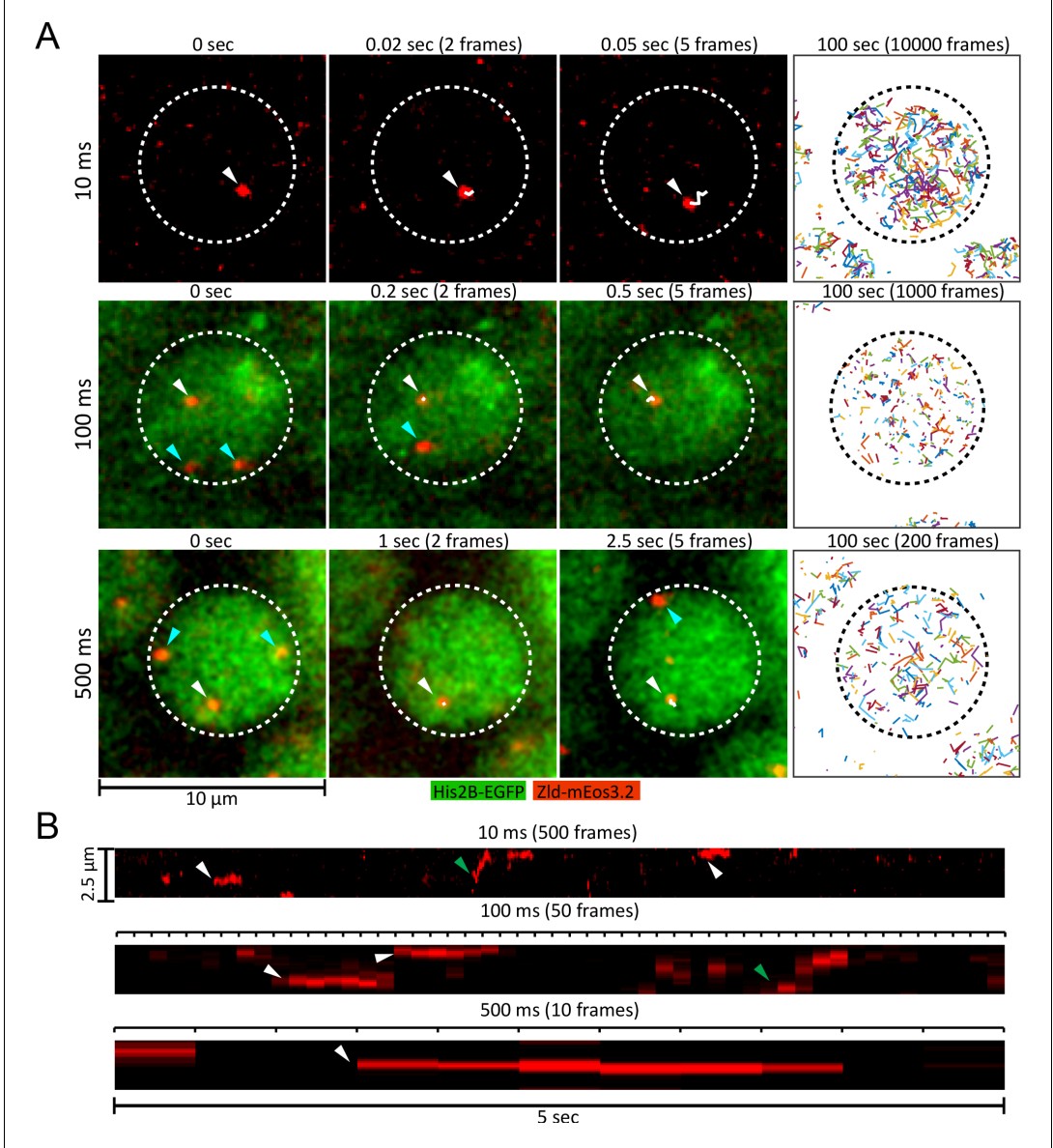

**Figure 1.** Live embryo single-molecule imaging and tracking of endogenous mEos3.2-Zld. (**A**) First three columns are example images showing single molecules of mEos3.2-Zld tracked over at least five frames (white arrows and trajectories) at frame rates of 10, 100 and 500 ms. Cyan arrows indicate molecules that appear for only one frame and are thus detected but not tracked. For the 100 and 500 ms data, enough signal is present in the His2B-eGFP channel from the 405 nm activation laser to enable simultaneous imaging of chromatin. Last column shows all single-molecule trajectories acquired in each nucleus over 100 s, corresponding to 539, 263, and 186 trajectories over 10000, 1000, and 200 frames for the 10, 100 and 500 ms data, respectively. Dotted lines indicate the boundary of a nucleus. Contrast was manually adjusted for visualization. (**B**) Representative kymographs over 5 s of imaging, corresponding to 500, 50, and 10 frames for the 10, 100 and 500 ms frame rate data, respectively. Green arrows point to molecules that display relatively large motions, and white arrows to immobile molecules.

DOI: https://doi.org/10.7554/eLife.40497.002

The following video and figure supplements are available for figure 1:

**Figure supplement 1.** Overview of CRISPR-Cas9 genome editing strategy.

DOI: https://doi.org/10.7554/eLife.40497.003

**Figure supplement 2.** Simplified Schematic of Lattice Light Sheet Microscope.

DOI: https://doi.org/10.7554/eLife.40497.004

**Figure supplement 3.** Mean detections per nucleus per frame for each frame rate.

DOI: https://doi.org/10.7554/eLife.40497.005

**Figure 1—video 1.** Movie illustrating ability to controllably photactiviate mEOS3.2.

*Figure 1 continued*

DOI: https://doi.org/10.7554/eLife.40497.006

**Figure 1—video 2.** Example movie of mEos3.2-Zld acquired at 10 ms frame rate.

DOI: https://doi.org/10.7554/eLife.40497.007

**Figure 1—video 3.** Example movie of mEos3.2-Zld (red) and His2B-EGFP (green) acquired at 100 ms frame rate.

DOI: https://doi.org/10.7554/eLife.40497.008

**Figure 1—video 4.** Example movie of mEos3.2-Zld (red) and His2B-EGFP (green) acquired at 500 ms frame rate.

DOI: https://doi.org/10.7554/eLife.40497.009

**Figure 1—video 5.** Example of a mobile molecule of mEos3.2-Zld tracked at 10 ms frame rate.

DOI: https://doi.org/10.7554/eLife.40497.010

**Figure 1—video 6.** Example of a immobile molecule of mEos3.2-Zld tracked at 10 ms frame rate.

DOI: https://doi.org/10.7554/eLife.40497.011

(*Hansen et al., 2017*; *Mir et al., 2017*; *Normanno et al., 2015*; *Chen et al., 2014a*; *Chen et al., 2014b*; *Mazza et al., 2012*). Thus, we use 10 ms data to measure the diffusion characteristics of immobile and mobile molecules, as well as to determine the fraction of total molecules that are immobile (bound) or mobile (*Hansen et al., 2018*), and 100 and 500 ms data to measure the duration and spatial distribution of binding events.

To gain an understanding of the dynamics of a protein which is stably associated with chromatin, we first examined single-molecule trajectories of the histone His2B at all three temporal scales. Histones are widely used as a benchmark for stably bound molecules (*Mazza et al., 2012*; *Hansen et al., 2017*; *Teves et al., 2016*), and we validate that His2B is a suitable control in the early *Drosophila* embryo through fluorescence recovery after photobleaching measurements (FRAP) (*Figure 2—figure supplement 1*). The FRAP data verifies that the majority of His2B molecules are bound for significantly longer times than Bcd or Zld, and indicates that the single-molecule trajectory lengths of His2B are limited by photobleaching and defocalization rather than unbinding. Consistent with this stable association, a visual examination of the single-particle trajectories of His2B at 10 ms frame rates illustrate that the vast majority of His2B molecules are immobile and confined within the localization accuracy of our measurements (*Figure 2*, top left and *Figure 2—video 1*). In comparison, the Zld and Bcd trajectories at 10 ms frame rates exhibit motions consistent with a mixed population of both chromatin-bound, slow diffusing, and mobile molecules (*Figure 2* left column and *Figure 2—video 1*).

When tracked over several seconds using exposure times of 100 and 500 ms (*Figure 2*, middle and right; *Figure 2—video 2* and *Figure 2—video 3*), the His2B trajectories now reflect the underlying motion of chromatin. We note a significantly greater apparent chromatin motion in early *Drosophila* embryos than is observed in mammalian cells in interphase where histones typically exhibit mobility less than the achievable localization accuracy (*Hansen et al., 2018*). At these slower frame rates, molecules of Zld and Bcd which are not immobile for a significant portion of the exposure time motion blur into the background (*Watanabe and Mitchison, 2002*; *Hansen et al., 2017*; *Zhang et al., 2012*). As a result, the trajectories of all three proteins now appear visually similar with the exception that His2B trajectories are longer in time due to their stable interaction with chromatin (*Figure 2—video 2* and *Figure 2—video 3*), with the length of trajectories now limited by unbinding, defocalization, and photobleaching. Having established His2B as a suitable control for a largely chromatin-bound protein, we next quantify and compare the single-molecule dynamics of Zld and Bcd in order to gain insight on how they explore the nucleoplasm and bind to DNA to regulate transcription.

## Bicoid and Zelda bind transiently and have large free populations

We first quantified the fraction of immobile molecules, and the diffusion coefficients of free and immobile molecules for His2B, Zld and Bcd, by analyzing the distributions of displacements (*Hansen et al., 2018*) from the high-speed (10 ms frame rate) data (*Figure 3A* and *Figure 3—figure supplement 1*). Visually the displacement distributions indicate that a greater fraction of both Zld and Bcd molecules are mobile (*Figure 3A*, displacements > 150–200 nm) than for His2B.

To quantify the single-molecule kinetics of all three proteins, the displacement distributions were fit to a two-state (free diffusing or immobile) kinetic model (*Figure 3—figure supplement 1*)

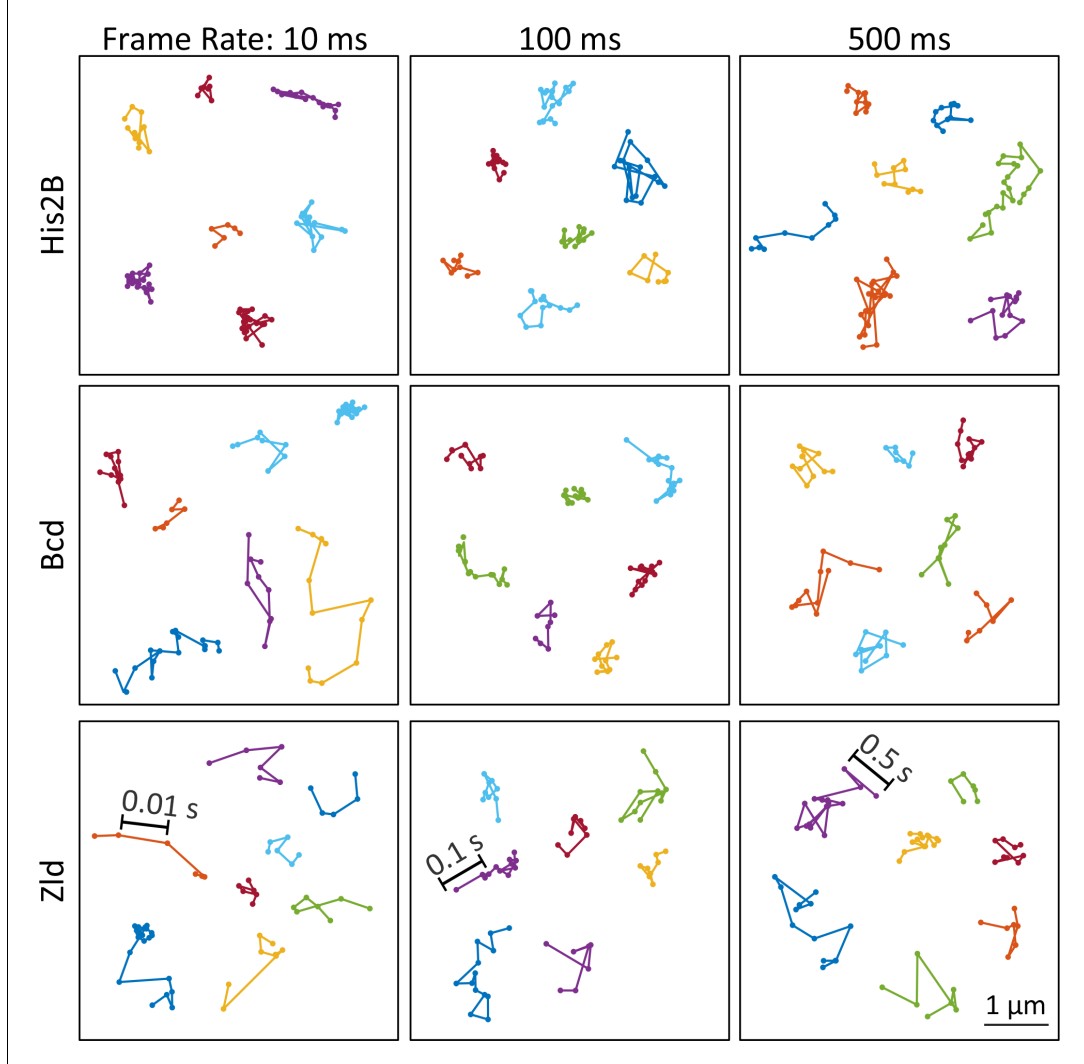

**Figure 2.** Representative single-molecule trajectories of His2B, Bcd and Zelda. Representative single-molecule trajectories of His2B, Bcd, and Zld from data acquired at frame rates of 10, 100 and 500 ms.

DOI: https://doi.org/10.7554/eLife.40497.012

The following video and figure supplement are available for figure 2:

**Figure supplement 1.** Fluorescence recovery after photobleaching (FRAP).

DOI: https://doi.org/10.7554/eLife.40497.013

**Figure 2—video 1.** Comparison of single-molecule movies for His2B-mEos3.2, mEos3.2-Bcd, and mEos3.2-Zld at 10 ms frame rate.

DOI: https://doi.org/10.7554/eLife.40497.014

**Figure 2—video 2.** Comparison of single-molecule movies for His2B-mEos3.2, mEos3.2-Bcd, and mEos3.2-Zld at 100 ms frame rate.

DOI: https://doi.org/10.7554/eLife.40497.015

**Figure 2—video 3.** Comparison of single-molecule movies for His2B-mEos3.2, mEos3.2-Bcd, and mEos3.2-Zld at 500 ms frame rate.

DOI: https://doi.org/10.7554/eLife.40497.016

assuming Brownian motion under steady-state conditions and taking into account effects from localization errors and defocalization bias (*Mazza et al., 2012*; *Hansen et al., 2017*; *Hansen et al., 2018*). While it is likely that the identified immobile and free diffusing populations contain more complex sub-populations, for example molecules exhibiting 1-D diffusion on DNA, a two-state model accurately fit the displacement distributions for all three proteins and using a higher number of states did not significantly improve the model fit to justify them. We find that ~50% of Zld and Bcd, and 88% of His2B molecules are immobile or bound (*Figure 3B* and *Figure 3—figure*

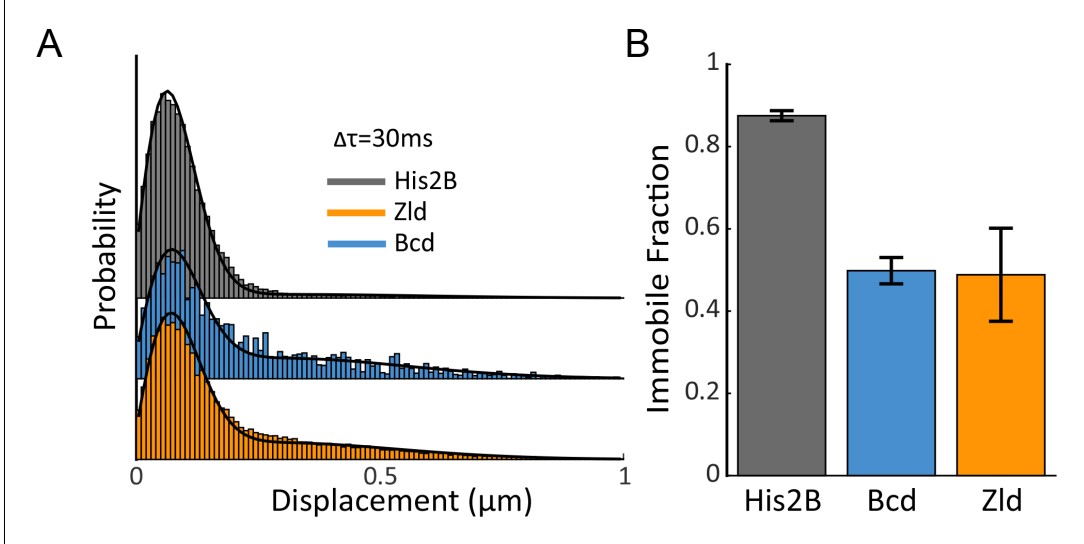

**Figure 3.** Immobile Fraction of His2B, Zld, and Bcd molecules. (A) Histograms of displacements for His2B, Zld, and Bcd after three consecutive frames (Δτ = 30 ms) at a frame rate of 10 ms. The Zld and Bcd distributions show a right tail indicative of a large free population that is missing from His2B distribution. Black lines are fits from two-state kinetic modelling, data shown is compiled from 3 embryos totalling 77869, 81660, and 11003 trajectories and 30, 128, 41 nuclei for His2B, Zld, and Bcd, respectively. (B) Fraction of molecules bound or immobile as determined from kinetic modeling of the displacement distributions, a summary of the model parameters is shown in *Figure 3—figure supplement 1*. Error bars show standard errors over three embryos.

DOI: https://doi.org/10.7554/eLife.40497.017

The following figure supplement is available for figure 3:

**Figure supplement 1.** Kinetic modeling of fast SPT data.

DOI: https://doi.org/10.7554/eLife.40497.018

*supplement 1*). The mean immobile or bound population diffusion coefficient for His2B is lowest followed by Bcd, and Zld, whereas the free diffusion coefficients for Zld are slightly lower than both Bcd, and His2B (*Figure 3—figure supplement 1*). The ~50% immobile population of Zld and Bcd indicate that both proteins spend roughly the same amount of time on nuclear exploration (searching for a binding target) and actually binding to chromatin (*Hansen et al., 2017*).

Next, we calculated the survival probability (the probability of trajectories lasting a certain amount of time) for the three factors at all three frame rates (*Figure 4A*). At all frame intervals, the length of His2B trajectories are, on average, longer than those of Zld and Bcd (*Figure 4A*). These longer trajectories reflect the greater fraction of bound His2B molecules as they defocalize with a lower probability. Since we expect, on average, the effects of nuclear and chromatin motion, as well as photobleaching, to be consistent for data acquired on the bound population of all three proteins, the longer His2B trajectories show both that His2B binds for longer than Bcd or Zld, and that unbinding and not photobleaching is likely to be dominant for Bcd and Zld trajectories at 500 ms exposure times, allowing us to estimate residence times.

To quantify genome average residence times, we fit the 500 ms survival probability distributions for Bcd, Zld, and His2B, to a two-exponential decay model (*Figure 4B*) to estimate the time constants associated with short- and long-binding events. As has been shown previously, the slow and fast time constants associated with the two exponents can be interpreted as the off-rates associated with non-specific and specific binding, respectively (*Hansen et al., 2017*; *Mir et al., 2017*; *Teves et al., 2016*; *Chen et al., 2014a*; *Chen et al., 2014b*). The resulting fits for Zld and Bcd are then bias corrected for photo-bleaching and defocalization using the fits to the His2B data. This correction is based on the fact that the long-lived population of His2B is associated with chromatin much longer than the dynamic range of our measurement time and that the maximum trajectory lengths that we measure for His2B are thus only limited by photo-bleaching and defocalization (see Materials and methods for more details) (*Hansen et al., 2017*).

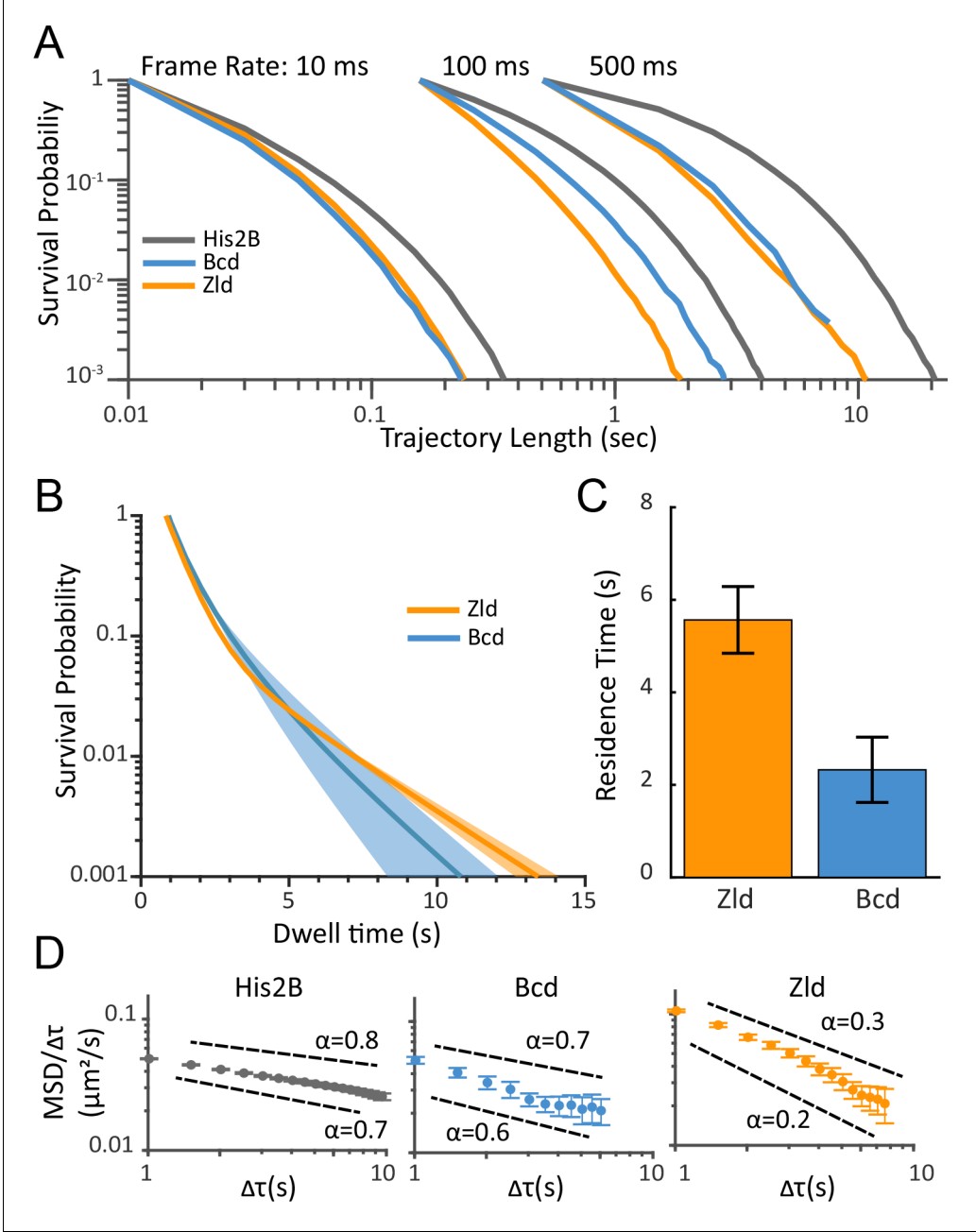

**Figure 4.** Residence times and dynamics of bound Zld and Bcd molecules. (**A**) Raw survival probabilities of trajectories at all frame rates. Calculated over 77869, 81660, 11003, at 10 ms, 107998, 42698, 8990 at 100 ms, and 2420, 14487, 47681 at 500 ms, trajectories for His2B, Zld, and Bcd respectively. (**B**) Uncorrected two-exponent fits to the survival probability distributions obtained from the 500 ms frame rate data. Dark solid lines are the mean over fits from three embryos and the shaded regions indicate the standard error. (**C**) Bias corrected quantification of the slow residence times for Zld ($5.56 \pm 0.72$ s) and Bcd ($2.33 \pm 0.71$ s). Error bars indicate standard error over three embryos for a total of 188 and 171 nuclei for Bcd and Zld, respectively. (**D**) MSD/$\tau$ curves for His2B, Bcd, and Zld at 500 ms frame rates plotted on log-log-scale. For anomalous diffusion MSD($\tau$)=$\Gamma\tau^{\alpha}$, where $\alpha$ is the anomalous diffusion coefficient. For MSD/$\tau$, in log-log space, the slope is thus 0 for completely free diffusion that is when $\alpha = 1$, and sub-diffusive($0<\alpha<1$), motions display higher negative slopes, with lower $\alpha$ corresponding to more anomalous motion.

DOI: https://doi.org/10.7554/eLife.40497.019

The following figure supplements are available for figure 4:

**Figure supplement 1.** Mean square displacement curves.

*Figure 4 continued on next page*

*Figure 4 continued*

DOI: https://doi.org/10.7554/eLife.40497.021

**Figure supplement 2.** Inference of residence times from single-molecule trajectories.

DOI: https://doi.org/10.7554/eLife.40497.020

Using this approach, we estimate genome average residence times for the specifically bound populations of ~5 s and ~2 s for Zld and Bcd, respectively (*Figure 4C*). This estimate for Bcd is slightly higher than we obtained previously (*Mir et al., 2017*), which we attribute to the more accurate bleaching correction using His2B here. These residence time estimates are consistent with FRAP measurements (*Figure 2—figure supplement 1*) where we measure recovery half times of ~5 and~1 s for Zld and Bcd, respectively.

Finally, prompted by a visual comparison of the Zld and Bcd trajectories with those of His2B and the relatively high diffusion coefficients for the bound or immobile population of all three proteins (*Figure 3—figure supplement 1*), we explore in more depth the kinetics of molecules that are relatively immobile to the extent that they don't motion blur into the background at 500 ms exposure times. We thus calculated and compared the time and ensemble averaged mean square displacement (TAMSD) of all three proteins (*Figure 4—figure supplement 1*). While TAMSD is not an appropriate metric for quantifying diffusion coefficients and bound fractions when the data contain a mixture of different dynamic populations such as at the 10 ms frame rate data (*Izeddin et al., 2014*; *Kepten et al., 2015*; *Hansen et al., 2018*), we reasoned that it is appropriate for a qualitative evaluation of the trajectories from the 500 ms data where we are measuring relatively stable immobile populations.

As expected for transcription factors and proteins confined within an environment, the TAMSDs for all three protein scale as $\sim\tau^{\alpha}$, where tau is the lag time and $\alpha$ is the anomalous exponent (*Normanno et al., 2015*; *Miné-Hattab et al., 2017*; *Izeddin et al., 2014*). To assess the level of anomalous motion, we plotted the TAMSD/$\tau$ curves from the 500 ms data for all three proteins in log-log scale (*Figure 4D*). Plotted in this manner a population of molecules exhibiting completely free diffusion would exhibit a log(TAMSD/$\tau$) curve of slope 0, that is when $\alpha = 1$, whereas sub-diffusive population ($0<\alpha<1$), display higher negative slopes, with lower $\alpha$ corresponding to more anomalous or confined motion. Our estimation of the $\alpha$ value for His2B is in agreement with previous measurements using Fluorescence Correlation Spectroscopy (*Bhattacharya et al., 2009*). Strikingly, while Bcd at 500 ms has a high $\alpha$ value similar to His2B, Zld has a low $\alpha$ value, consistent with a high amount of anomalous motion (*Figure 4D*).

Anomalous or sub-diffusive motion can result from a range of underlying physical interactions including aggregation, weak interactions with other proteins and chromatin, repetitive binding at proximal binding sites, among many other possibilities (*Woringer and Darzacq, 2018*; *Fradin, 2017*). The complexity of the TAMSDs from Zelda trajectories acquired at 500 ms suggest that at these frame rates we likely measure a mixture of effects that lead to a relatively immobile population of Zelda. Given that Zld is known to exhibit an extremely heterogeneous sub-nuclear spatial distribution, we next examined the bulk rather than single-molecule spatial-temporal dynamics of Zld.

## Zelda and Bicoid form dynamic subnuclear hubs

Recently, a highly clustered spatial distribution of Zld was reported (*Dufourt et al., 2018*), but the temporal dynamics of these clusters have not been examined due to the technical limitations of confocal microscopy. We thus performed high-resolution 4D imaging using LLSM of Zld in developing embryos. We find that the spatial distribution of Zelda is highly dynamic and linked to the nuclear cycle (*Figure 5—video 1*). We observe that Zld rapidly loads into nuclei near the end of telophase and associates to the still condensed chromatin. As the chromatin de-condenses and the nuclei enter interphase, Zld breaks into smaller highly dynamic clusters (*Figure 5A* and *Figure 5—video 2*). As the nucleus enters prophase and the nuclear membrane begins to break up, Zld appears to leave the nucleus and correspondingly the cytoplasmic signal around the nucleus increases (*Figure 5—video 1*). As the Zld concentration around the chromatin drops, so does the appearance of clusters, although Zld appears to remain associated with chromatin until the end of prophase. From the end

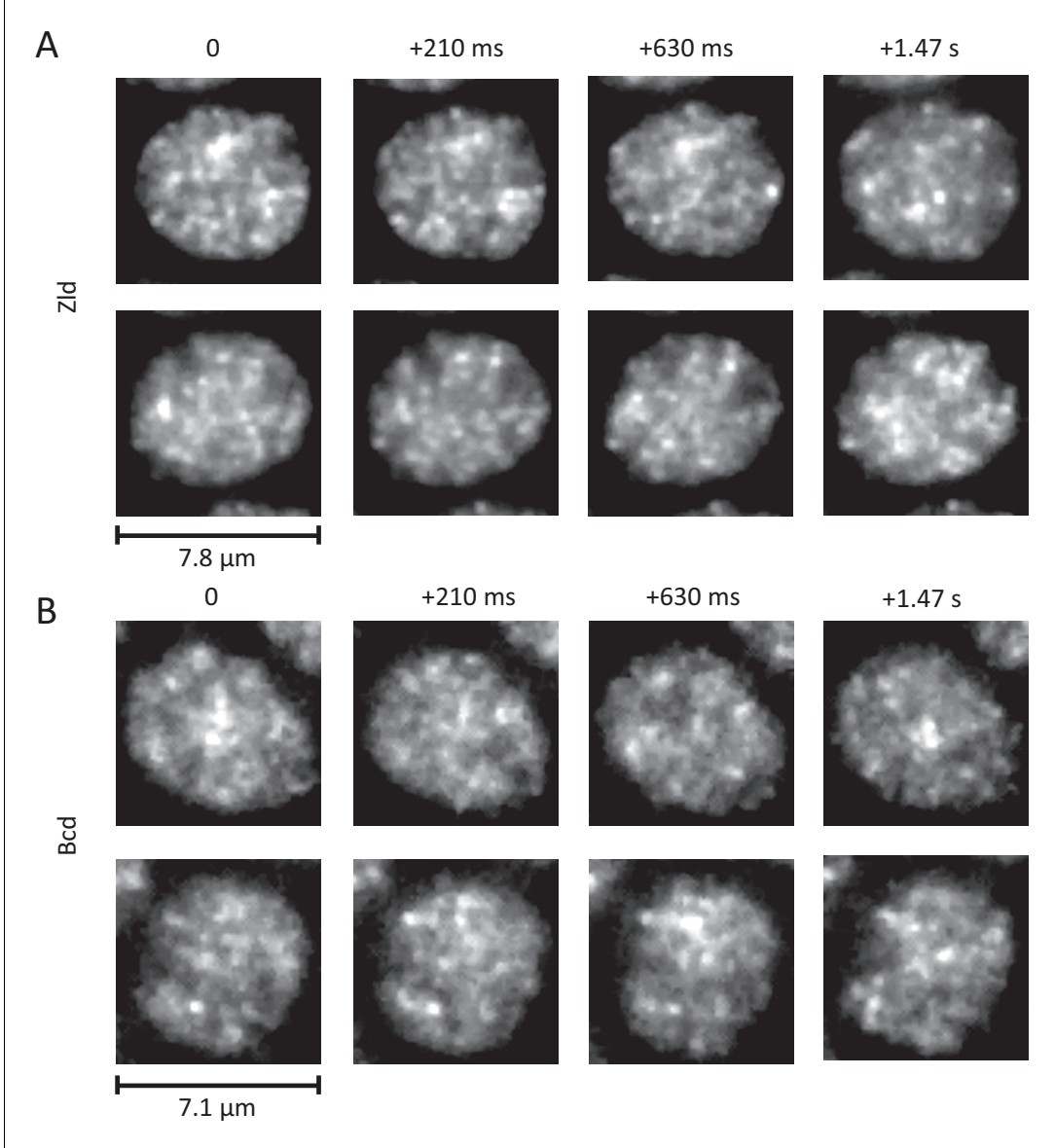

**Figure 5.** Dynamic interphase hubs of Zld and Bcd. Example images of the spatial distributions of Zld (**A**) and Bcd (**B**) at various time intervals illustrating the dynamic nature and wide range of size distributions and temporal persistences of enriched hubs (see also *Figure 5—videos 2–4*). mNeonGreen-Zld and EGFP-Bcd were imaged at 15 ms and 210 ms frame rates, respectively. To allow comparison, the sum projection of 14 frames (210 ms total integration) is shown for Zld. Images were processed with a 1-pixel radius median filter to remove salt-and-pepper noise and contrast-adjusted manually for visual presentation.

DOI: https://doi.org/10.7554/eLife.40497.022

The following videos are available for figure 5:

**Figure 5—video 1.** Cell cycle dynamics of Zelda spatial distribution.
DOI: https://doi.org/10.7554/eLife.40497.023

**Figure 5—video 2.** Four-dimensional Interphase dynamics of Zld spatial distributions.
DOI: https://doi.org/10.7554/eLife.40497.024

**Figure 5—video 3.** Interphase dynamics of Zld spatial distributions at high temporal resolution.
DOI: https://doi.org/10.7554/eLife.40497.025

**Figure 5—video 4.** Four dimensional dynamics of Bcd spatial distribution.
DOI: https://doi.org/10.7554/eLife.40497.026

of prophase to telophase, no Zld is observed in proximity of the condensed chromatin until it rapidly loads back in to reforming nuclei at the end of telophase.

The dynamic interphase hubs of Zld are amorphous, appear to have a wide distribution of sizes, and persist for highly variable amounts of time (*Figure 5A* and *Figure 5—video 2*). When imaged at higher temporal resolutions (*Figure 5—video 3*) we observe that both hub location and intensity vary even at sub-second time-scales, suggesting that there is dynamic exchange of Zld molecules in clusters with the rest of the nucleoplasm. Bulk imaging of Bcd also reveals that it forms dynamic hubs in interphase (*Figure 5B* and *Figure 5—video 4*) although they appear less prominent both in size and temporal persistence than those of Zld. Our observations of the nuclear cycle dynamics of Bcd are consistent with previous reports of it filling into the nucleus after mitosis (slower than Zld) and a slow decrease in concentration after the nuclear membrane breaks down (*Gregor et al., 2007*).

These observations of highly heterogeneous and dynamic sub-nuclear distributions are consistent with our earlier work where we observed that Bcd binding is clustered in discrete subnuclear hubs (*Mir et al., 2017*). We also previously showed that these Bcd hubs do not form in the absence of maternal Zld which naturally led us to next ask whether there is a relationship between Bcd binding and the local concentration of Zld.

## Bicoid binding events are enriched in Zelda hubs

To explore the relationship between Zld hubs and Bcd binding, we performed dual-color experiments recording the single-molecule dynamics of Bcd using mEos3.2-Bcd and the bulk spatial distribution of Zld using mNeonGreen-Zld. To strike a balance between the constraints of the imaging system, the dynamic range of the single-molecule trajectories, and the fast dynamics of Zld hubs (*Figure 5* and *Figure 6A*), we acquired a bulk fluorescence image of mNeonGreen-Zld with a 1 s acquisition time followed by 10 frames of single-molecule imaging of mEos3.2-Bcd with a frame rate of 100 ms (*Figure 6—video 1*).

Using the bulk Zld data, we partitioned nuclei into regions of high- and low- relative Zld density (*Figure 6B–C* and *Figure 6—figure supplement 1*), where high-density regions correspond to Zld hubs. Parsing the Bcd single-molecule data, we find that the enrichment of bound Bcd molecules is consistently higher within the high Zld density regions (*Figure 6D*). In the embryo anterior, where Bcd concentrations are highest, there is a two-fold increase in the enrichment of Bcd trajectories in high-density Zld regions compared to the rest of the nucleoplasm. Along the anteroposterior axis, the enrichment of Bcd trajectories within the high-density Zld regions increases, to an excess of around four-fold in the posterior (*Figure 6D*), whereas the Zld spatial distribution remains unchanged. This observation is consistent with our previous report of more pronounced clustering of Bcd in the embryo posterior (*Mir et al., 2017*). When we examined the stability of Bcd binding as a function of relative Zld density, we also find that at more posterior embryonic positions longer binding events of Bcd are associated with higher Zld density, in contrast to the embryo anterior (*Figure 6—figure supplement 2*). We note that while there is an increase in long Bcd-binding events in Zld hubs this effect is not large enough to account for the overall enrichment of all Bcd binding events, suggesting that Zld increases the time averaged Bcd occupancy at DNA-binding sites by increasing its local concentration (increasing $k_{on}$) and not by increasing its residence times at its target sites (decreasing $k_{off}$).

This association between Zld hubs and Bcd binding suggests that these hubs, although dynamic and transient, might be preferentially forming on genes that are co-regulated by Zld and Bcd. Furthermore, given the strong association of Zld binding measured by chromatin immunoprecipitation with the binding of many early embryonic factors (*Harrison et al., 2011*) we expected a strong correlation between Zld hubs and sites of active Bcd-dependent transcription. To test this hypothesis, we next performed imaging of the spatial distributions of Bcd and Zld in the context of active transcription.

## Zelda hubs are not stably associated with sites of active transcription

We chose to study the relationship between Zld and Bcd hubs and transcriptional activity at the canonical Bcd target gene *hunchback* (*hb*). The *hb* gene was the first identified target of Bcd (*Struhl et al., 1989*; *Tautz, 1988*), and its anterior transcription is dramatically disrupted in the

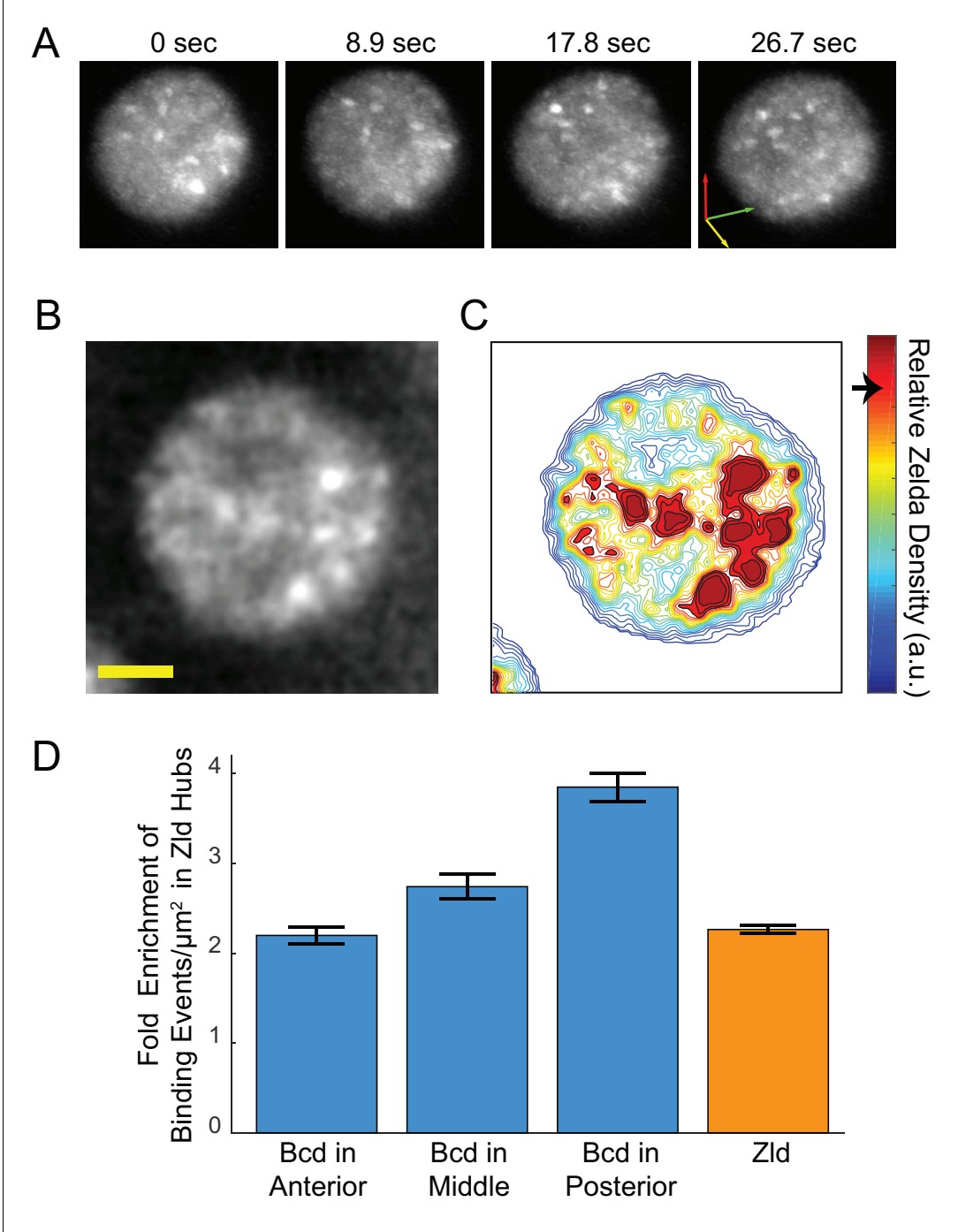

**Figure 6.** Enrichment of Bcd binding in Zld hubs. (**A**) Three-dimensional volume renderings of an interphase nucleus showing the dynamic nature of Zld hubs (see *Figure 5—video 2*, *Figure 5—video 3*).The 3D axes indicate the xyz axes and the arrow lengths are 2 μm along each direction. (**B**) Representative snapshot of the interphase distribution of Zelda, yellow scale bar is 2 μm. (**C**) Relative Zelda Density map for the nucleus shown in (**B**), the arrow on the colorbar indicates the threshold for defining a region as high density (corresponding to hubs). (**D**) Fold enrichment in the density of bound molecules of Bcd in the anterior, middle, and posterior embryo, and of Zld, in Zld hubs vs. the rest of the nucleoplasm. Error bars show standard error over three embryos with a total of 1344, 3921, 481 nuclear images for Bcd Ant, Mid, and Post, respectively, and 4399 for Zld.
DOI: https://doi.org/10.7554/eLife.40497.027

The following video and figure supplements are available for figure 6:

*Figure 6 continued on next page*

*Figure 6 continued*

**Figure supplement 1.** Analysis of Zelda density.
DOI: https://doi.org/10.7554/eLife.40497.028
**Figure supplement 2.** Cumulative probability of trajectories vs. relative Zelda density.
DOI: https://doi.org/10.7554/eLife.40497.029
**Figure 6—video 1.** Bcd single-molecule localizations in context of the bulk spatial distribution of Zld.
DOI: https://doi.org/10.7554/eLife.40497.030

---

absence of Bcd protein (*Staller et al., 2015*; *Ochoa-Espinosa et al., 2009*; *Hannon et al., 2017*). The regulatory sequences for *hb* contain multiple clustered Bcd-binding sites, as well as recognizable Zelda motifs (*Harrison et al., 2011*) (*Figure 7—figure supplement 1*). ChIP studies show that both Zelda and Bcd bind strongly at the *hb* locus, though loss of Zelda has only a modest quantitative effect on *hb* expression (*Combs and Eisen, 2017*; *Nien et al., 2011*). An enrichment of Bcd in the vicinity of active *hb* loci was previously observed using FISH (*Xu et al., 2015*) on fixed embryos, but nothing is known about the dynamics of this enrichment or its relationship to Zld.

To visualize the *hb* locus, we took advantage of the MS2 system, which allows fluorescent labelling of nascent transcripts of specific genes (*Garcia et al., 2013*; *Bothma et al., 2015*). Bothma, et al. generated a fly line carrying a bacterial artificial chromosome (BAC) that contains an MS2-labeled *hb* locus that closely recapitulates the expression of *hb* itself (*Bothma et al., 2015*). We thus performed high spatio-temporal resolution 4D imaging of the bulk distributions of Zld, and separately Bcd, in embryos carrying the *hb* BAC and MCP-mCherry (*Figure 7A*, *Figure 7—figure supplement 1*, *Figure 7—video 1*, *Figure 7—video 2*, *Figure 7—video 3*).

From visual examination of movies of Zld and Bcd in the presence of *hb* transcription (*Figure 7—video 2*, *Figure 7—video 3* and *Figure 7B*), we observe that the temporal relationship between Zld and Bcd hubs and the *hb* locus in nuclei where it is expressed (and therefore visible) is highly dynamic. We do not observe stable associations between high-concentration hubs of either factor and *hb*. However, we do see that contacts between *hb* and hubs of both factors occur frequently, so we next asked whether *hb* showed any preferential association with hubs of either factor over time.

Following *Spiluttini et al. (2010)*, we averaged the Bcd and Zld signal surrounding active *hb* loci over thousands of images from six embryos (*Figure 7C*, *Figure 7—video 4*, *Figure 7—video 5*). For Bcd, we observe a sharp enrichment of fluorescent signal at the *hb* locus in comparison to randomly selected control points within nuclei (*Figure 7D*). We observe no such enrichment for Zld at *hb* (*Figure 7B*), however we note that Zld has many fold more targets than Bcd (*Harrison et al., 2011*; *Li et al., 2008*), and its target loci may be at too high a density in the nucleoplasm to detect enrichment at any one of them with this assay. These results imply a previously unappreciated aspect of the relationship between transcription factor hubs and their target genes: that individual hubs are multifactorial and likely service many different genes and loci within the nucleus.

## Discussion

We previously reported a strong correlation between genomic locations of Zld and Bcd binding (*Harrison et al., 2011*) and suggested, based on a roughly twenty-fold increase in the occupancy of potential Bcd-binding sites in Zld bound regions, that there is strong cooperativity between these two factors. More recently, we showed that Bcd binds DNA highly transiently, but that its binding is spatially organized in a Zld-dependent manner (*Mir et al., 2017*).

Based on these data, we hypothesized that Zld could act as a DNA-bound scaffold facilitating Bcd binding by increasing its local concentration in the vicinity of its target. Here, however, we find that Zld also binds DNA transiently and therefore cannot, by itself, act as a stable scaffold at enhancers.

Our observation that Zld and Bcd form hubs of locally high concentration suggests an alternative model in which multiple factors enriched within hubs interact to increase factor occupancy at DNA targets without the need for stabilizing this interaction by conventional 'lock and key' interactions. In our model, efficient occupancy of a site is achieved by frequent transient weak binding events within hubs rather than long, stable interactions on DNA (*Woringer and Darzacq, 2018*).

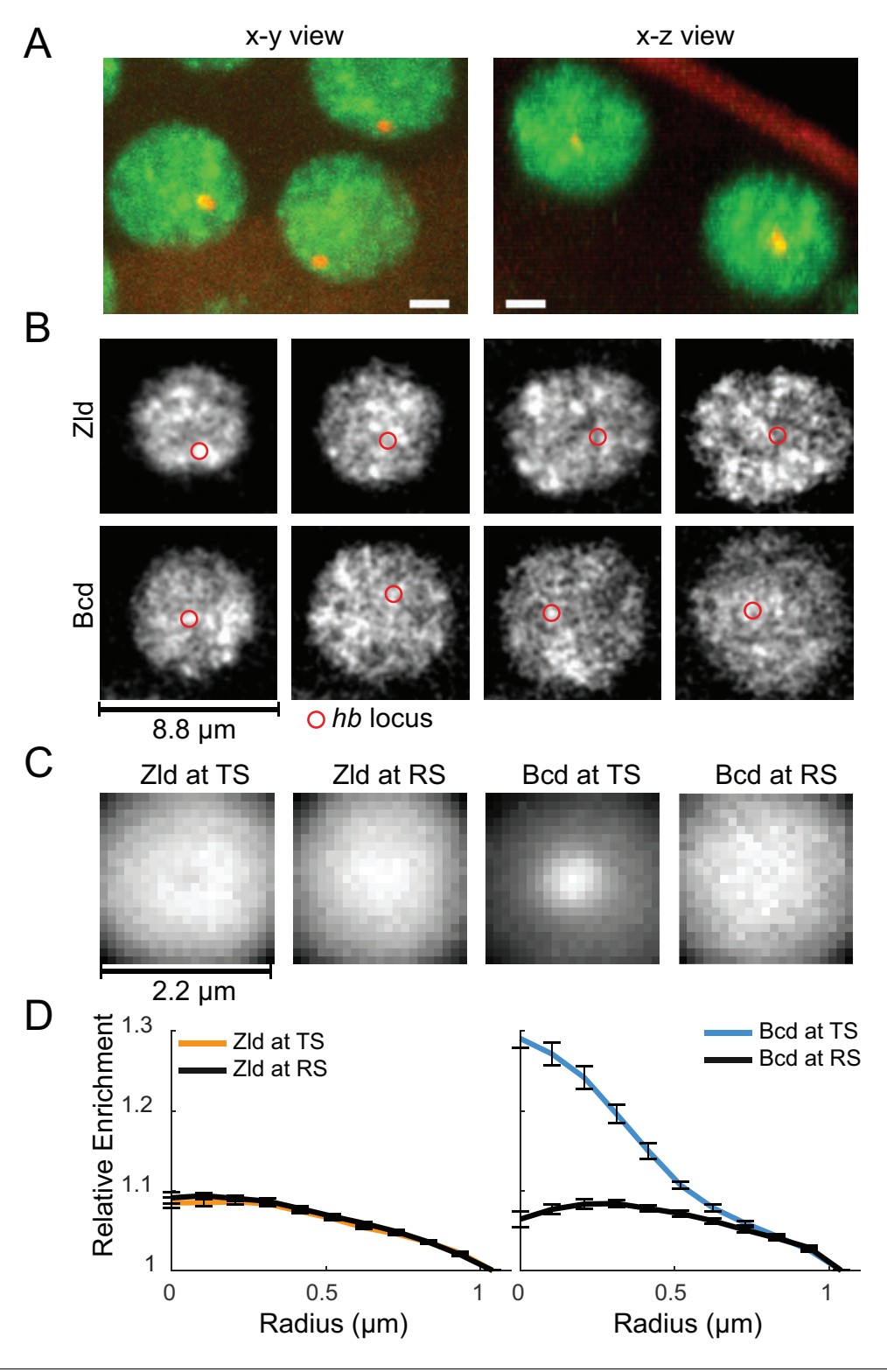

**Figure 7.** Spatio-temporal distribution of Zld and Bcd hubs in context of active *hb* loci. (**A**) Representative x-y and x-z max projections over a nuclear diameter of mNeonGreen-Zld (green) and an active hb locus tagged with MS2-MCP-mCherry (red) white scale bars are 2 μm. (**B**) Representative snapshots of the distribution of Zld and Bcd with the *hb* locus indicated by the red circle. Images suggest that high concentration Bcd hubs frequent the active locus whereas Zld exhibits more transient and peripheral interactions. Contrast of each image was manually

*Figure 7 continued on next page*

*Figure 7 continued*

adjusted for visualization and comparison. (**C**) Average Zld and Bcd signals in a 2.2 μm window centered at active *hb* locus (TS) and at random sites in the nucleus (RS). Averages were calculated over 3943 and 6307 images of active loci from six embryos for Bcd and Zld, respectively (see *Figure 7—videos 4* and *5*). (**D**) Radial profiles of the images in C, normalized to one at the largest radius. Error bars show standard error over all images analyzed.
DOI: https://doi.org/10.7554/eLife.40497.031

The following video and figure supplement are available for figure 7:

**Figure supplement 1.** MS2 system and data analysis.
DOI: https://doi.org/10.7554/eLife.40497.032
**Figure 7—video 1.** Four dimensional imaging of protein distribution in the context of transcription.
DOI: https://doi.org/10.7554/eLife.40497.033
**Figure 7—video 2.** Example of Bcd spatial distribution around an active *hb* locus.
DOI: https://doi.org/10.7554/eLife.40497.034
**Figure 7—video 3.** Example of Zld spatial distribution around an active *hb* locus.
DOI: https://doi.org/10.7554/eLife.40497.035
**Figure 7—video 4.** Calculation of average Zld signal around active *hb* loci.
DOI: https://doi.org/10.7554/eLife.40497.036
**Figure 7—video 5.** Calculation of average Bcd signal around active *hb* loci.
DOI: https://doi.org/10.7554/eLife.40497.037

A strong correlation between transcription and hubs of RNA polymerase II (*Cisse et al., 2013*; *Chong et al., 2018*; *Boehning et al., 2018*) and transcription factors (*Chen et al., 2014a*; *Chen et al., 2014b*; *Chong et al., 2018*; *Wollman et al., 2017*; *Liu et al., 2014*) has been reported in mammalian cells. In *Drosophila*, high local concentrations of the transcription factor Ultrabithorax (Ubx) at sites of Ubx-mediated transcription has recently been reported (*Tsai et al., 2017*).

However, use of LLSM to image hubs at high frame rates (*Figure 5*, *Figure 5—video 3*) shows that they are not stable structures, and furthermore, that they interact only transiently with sites of active transcription. We suggest that what previous reports (*Dufourt et al., 2018*) have described as stable clusters are actually time-averaged accumulations (such as what we observe for Bcd at *hb*) resulting from insufficient temporal resolution or use of immunofluorescence data, rather than discrete sub-nuclear bodies.

The most surprising observation we report is that at an active locus, transcription occurs with rare and transient visits of hubs containing the primary activator of the locus. One possible explanation is some version of the decades old 'hit-and-run' model proposed by *Schaffner (1988)* in which transcription factor binding and interactions with enhancers are only required to switch promoters into an active state, after which multiple rounds of transcription could occur in the absence of transcription factor binding (*Para et al., 2014*; *Doidy et al., 2016*). Transcription could then be regulated by the frequency of visits, rather than stable association.

Our observation that hubs form rapidly at the exit from mitosis and are most pronounced at times when no transcription happens raises the possibility of an even greater temporal disconnect. We and others have suggested that a primary role of Zld is in the licensing of enhancer and promoter chromatin for the binding of other factors (*Li et al., 2014*; *Foo et al., 2014*; *Schulz et al., 2015*; *Sun et al., 2015*) and in the creation and stabilization of chromatin three-dimensional structure (*Hug et al., 2017*). It is possible that the key point of activity occurs early during each nuclear cycle when the chromatin topology is challenged by the replication machinery and transcription has not begun (*Blythe and Wieschaus, 2016*).

Our single-molecule data point to intrinsic forces that might lead to the formation and maintenance of Zld hubs. The highly anomalous nature of the movement of Zld that differs from that of chromatin associated His2B and Bcd suggests that its motion depends at least in part on interactions off of DNA. Most of the amino acid sequence of Zld consists of intrinsically disordered domains, some of which are required for its function (*Hamm et al., 2017*). We and others have shown that intrinsically disordered domains mediate weak, multivalent protein-protein interactions between regulatory factors (*Chong et al., 2018*; *Boehning et al., 2018*; *Kovar, 2011*; *Burke et al., 2015*; *Altmeyer et al., 2015*; *Xiang et al., 2015*; *Friedman et al., 2007*) that lead to the selective enrichment of factors in hubs (*Chong et al., 2018*; *Boehning et al., 2018*). We think it is therefore highly

likely that Zld, and therefore Bcd, hubs are formed by interactions involving intrinsically disordered domains.

Bcd is only one of many proteins whose early embryonic binding sites have a high degree of overlap with those of Zld (*Harrison et al., 2011*), including many other factors involved in anterior-posterior and dorsal-ventral patterning, and other processes (*Reichardt et al., 2018*; *Foo et al., 2014*; *Sun et al., 2015*; *Schulz et al., 2015*; *Nien et al., 2011*; *Xu et al., 2014*; *Pearson et al., 2012*; *Boija and Mannervik, 2016*; *Shin and Hong, 2016*; *Ozdemir et al., 2014*). We hypothesize that Zld provides scaffolds to form distinct hubs with each of these factors, mediated by a combination of weak and transient protein:protein and protein:DNA interactions.

Such hubs could help explain the disconnect between the canonical view of transcription factor function based on their directly mediating interactions between DNA and the core transcriptional machinery and data like those presented here that suggests a more stochastic temporal relationship. There is abundant evidence that transcription factors can affect promoter activity by recruiting additional transcription factors, chromatin remodelers and modifiers, and other proteins. Extrapolating from our observation that Zld appears to form some type of scaffold for Bcd hubs, we propose that hubs contain not only transcription factors, but loose assemblages of multiple proteins with diverse activities. Such multifactor hubs could provide each transcription factor with a bespoke proteome, with far greater regulatory capacity and precision than could plausibly be achieved through stable direct protein-protein interactions involving each factor.

# Materials and methods

## Key resources table

| Reagent type (species) or resource | Designation | Source or reference | Identifiers | Additional information |
|---|---|---|---|---|
| Strain, strain background (*D. melanogaster*) | mNeonGreen-Zld | this paper | | Fly line with an N-terminal mNeonGreen fusion tag inserted at the endogenous Zld locus. |
| Strain, strain background (*D. melanogaster*) | mEos3.2-Zld | this paper | | Fly line with an N-terminal mEos3.2 fusion tag inserted at the endogenous Zld locus. |
| Strain, strain background (*D. melanogaster*) | H2B-EGFP | this paper | | Fly line with an H2B-EGFP transgene inserted on chromosome 3. Transgene is expressed ubiquitously under the control of a synthetic tubulin promoter. |
| Strain, strain background (*D. melanogaster*) | H2B-mEos3.2 | this paper | | Fly line with an H2B-mEos3.2 transgene inserted on chromosome 3. Transgene is expressed ubiquitously under the control of a synthetic tubulin promoter. |
| Strain, strain background (*D. melanogaster*) | MCP-mcherry | H Garcia lab | | Fly line with MCP-mCherry inserted as a transgene on chromosome 2. |
| Genetic reagent (*D. melanogaster*) | sgRNA #1 bicoid | this paper | | sgRNA targeting N-terminus of Bcd gene, sequence GCGGAGTG TTTGGGGAAAA |

*Continued on next page*

*Continued*

| Reagent type (species) or resource | Designation | Source or reference | Identifiers | Additional information |
|---|---|---|---|---|
| Genetic reagent (*D. melanogaster*) | sgRNA #1 bicoid | this paper | | sgRNA targeting N-terminus of Bcd gene, sequence TAAAAGTTT TGATCTGGCGG |
| Genetic reagent (*D. melanogaster*) | sgRNA #1 bicoid | this paper | | sgRNA targeting N-terminus of Bcd gene, sequence TGATGGTAAA AGTTTTGATC |
| Genetic reagent (*D. melanogaster*) | sgRNA Zelda | M Harrison lab | | sgRNA targeting N-terminus of Zld gene, sequence CCTCTGCCGC GTGCAGGGG |
| Software, algorithm | Spot-On | *Hansen et al., 2018*, eLife | | |

## Generation of transgenic fly lines

The following fly lines were constructed using CRISPR/Cas9 mutagenesis with homology directed repair: mNeonGreen-Zelda, mEos3.2-Zelda, mEos3.2-Bicoid. sgRNAs targeting sites near the desired insertion sites were cloned via the primer annealing method into plasmid pMRS-1, which is a version of pCFD3 (*Port et al., 2014*) (addgene #49410) with alterations to the sgRNA body made according to (*Chen et al., 2013*). sgRNA sequences were CCTCTGCCGCGTGCAGGGG for Zelda and an equimolar mixture of TGATGGTAAAAGTTTTGATC, GCGGAGTGTTTGGGGAAAA, and TAAAAGTTTTGATCTGGCGG for Bicoid. Homology directed repair templates were constructed in a pUC19 backbone via Gibson assembly with the desired tag, with an N-terminal FLAG tag, flanked by 1 kb homology arms. Fluorescent tags were inserted at the ATG located at 3R:6759268 for Bicoid and at X:19782283 for Zelda (dm6 coordinates). For both Zelda and Bicoid, the natural start ATG was removed, and the fluorescent protein and linker were fused with the first amino acid after the initiation Methionine residue. We tested a number of linker sequences and found variable tag- and protein-specific effects on viability. Linker sequences that yielded homozygous viable animals were GDGAGLIN (mNeonGreen-Zld), GGGGSGSGGS (mEos3.2-Zld and mEos3.2-Bcd) and GGGGSGSGG SMTRDYKDDDDKTRGS (H2B-mEos3.2 and H2B-EGFP).

HDR template and sgRNA plasmids were sent to Rainbow Transgenic Flies, Inc. (Camarillo, CA) to be injected into embryos expressing Cas9 in the germline. Resulting adult flies were crossed to flies possessing balancer chromosomes matching the relevant chromosome. Single F1 progeny carrying the marked balancer were crossed to balancer stock flies, allowed 4–8 days for females to lay sufficient eggs, and the F1 parents were sacrificed for PCR genotyping. Typically, we find that ∼ 10–40% of F1 animals contain insertions. For example, for a pooled injection of three Zelda tagging constructs, we screened 79 F1 animals and recovered 36 hits, of which we screened 19 to determine the identify of the inserted tag, recovering 8 GFP-Zld, 6 mEos3.2-Zld, and 5 HaloTag-Zld. We find that efficiency varies for different genes and injections, but these results are fairly typical. For positive hits, balanced lines were generated by selecting appropriately-marked F2 progeny, and F3 animals were examined for the presence of homozygous animals, revealed by the lack of balancer phenotype. As Bicoid has only maternal phenotypes, homozygous mothers were tested for the ability to give viable offspring. Lines that tolerated homozygous insertions were subjected to further screening by the preparation of clean genomic DNA, amplification of the locus using primers outside the donor homology arms, and subsequent Sanger sequencing of the entire amplicon. Lines carrying insertions free of mutations and containing no incorporated plasmid backbone were kept and utilized for imaging experiments.

His2B-mEos3.2 was introduced as a supplemental transgene under the control of a ubiquitous pΔTubHA4C promoter (*Zhang et al., 2013*) and SV40 3' UTR via PhiC31-mediated recombinase

(*Groth et al., 2004*) into landing site VK33 (*Venken et al., 2009*). A transgene was used to avoid potential complications associated with editing the highly multicopy histone locus.

We chose a red fluorescent protein for single-molecule imaging in embryos as better signals are achievable at longer wavelengths. First, as is well known, there is high autofluorescence at greener wavelengths in the *Drosophila* embryo (and for most biological materials). Second, Rayleigh scattering, scattering from particles of sizes less than the wavelength of the imaging light (the phenomenon responsible for blue skies and red sunsets), scales as $\sim 1/\lambda^4$ where $\lambda$ is the imaging wavelength. Thus, using longer wavelengths results in fewer photons being scattered and thus more photons being absorbed, emitted and collected from single molecules (*Mir et al., 2018*).

## Western blot

For each genotype indicated, 50–60 embryos aged 2 hr at 25° were dechorionated in bleach, rinsed in salt solution (NaCl with TritonX-100) and flash frozen. Frozen embryos were homogenized in 50 µL of sample buffer and 20 µL of sample per lane was loaded onto two separate 4–15% SDS-PAGE gels (Bio-Rad Cat # 4561083DC). Western blots were performed using rabbit polyclonal α-Bcd or α-Zld primary antibodies and a goat α-rabbit HRP conjugated secondary antibody (Thermo Fisher Cat # 31460).

## MS2 crosses

For MS2 experiments, yw; +; MCP-mCherry (gift from S. Alamos and H.G. Garcia) virgin females were crossed to males homozygous for either EGFP-Bcd or mNeonGreen-Zld. Resulting female progeny maternally deposit both MCP and the labeled TF in embryos. Virgin females were crossed to males homozygous for the *hb* MS2 BAC and resulting embryos were used for imaging.

## Lattice light-sheet microscopy of live embryos

Embryos were collected from flies in small cages over a 90-min laying period. Prior to embryo collection, the surface of a 5 mm diameter glass coverslip was made adhesive by deposition of of a small drop of glue solution (the glue solution was prepared by dissolving a roll of double-sided scotch tape in heptane overnight). The coverslip was allowed to dry for at least 5 min, which is sufficient time for the heptane to evaporate leaving behind a sticky surface. Embryos were washed off from the cage lids using tap water and gentle agitation with a paintbrush into a nylon cell-strainer basket. Embryos were then dechorionated in 100% bleach for 90 s. The dechorionation was then stopped by continuous washing under tap water until no further bleach smell could be detected, typically 30 s. The embryos were then transferred from the water filled strainer basket onto an agar pad using a fine haired paintbrush and arranged into an array of typically 3 rows and five columns with a consistent anteroposterior (A-P) orientation. The arranged embryos were then gently contact transferred onto the adhesive coverslip which was subsequently loaded into the microscope sample holder.

A home built lattice light-sheet microscope (LLSM) was used (*Chen et al., 2014a*; *Chen et al., 2014b*; *Mir et al., 2017*) for all single molecule, bulk fluorescence, and MS2 imaging experiments. Images were acquired using two Hamamatsu ORCA-Flash 4.0 digital CMOS cameras (C13440-20CU). An image splitting long-pass dichroic (Semrock FF-560) was placed in between the two cameras to separate emission wavelengths of over and under 560 nm, in addition bandpass filters corresponding to the fluorophore of interest were installed in front of each camera to provide further spectral filtering (Semrock FF01-525/50 for mNeon and sfGFP, Semrock FF01-593/46 for mEOS3.2, and Semrock FF01-629/53 for mCherry). Further details of imaging settings and conditions for each type of imaging experiment are provided in the corresponding sections below.

For all experiments, the stage positions corresponding to the anterior and posterior extents of each embryo imaged were recorded. The position along the anteroposterior axis for each image or movie recorded was then calculated as a fraction of the embryonic length (EL) with 0 and 1 to the anterior and posterior extents of the embryo, respectively. The nuclear cycle and progression within the nuclear cycle (e.g. interphase, prophase, mitosis) were also recorded for each movie or image. Times between nuclear cycles were also monitored to ensure that data was being acquired on a healthy and normally developing embryos. Embryos which exhibited aberrant development, for example longer than usual nuclear cycles, or numerous aberrant nuclear divisions were abandoned and the data was discarded.

## Single-molecule imaging and tracking in live embryos

For single-molecule imaging experiments, the illumination module of the LLSM was modified to provide constant photoactivation using a 405 nm laser line that bypasses the Acousto-optical tunable filter (AOTF) (*Figure 1—figure supplement 1*). We found that even when a lattice pattern for 561 nm was displayed on the spatial light modulator (SLM) sufficient 405 nm illumination was present in the imaging plane to allow for controlled photo-activation of mEos3.2-Bcd and mEos3.2-Zld. For all single-molecule experiments, a 30 beam square lattice with 0.55 and 0.44 inner and outer Numerical Apertures, respectively, was used in dithered mode for excitation. The 405 nm laser line was kept on constantly during the acquisition period for photoswitching and a 561 nm laser line was used for excitation. For both mEos3.2-Bcd and mEos3.2-Zld, data was acquired at 7.5, 100 and 500 ms exposure times with effective frame rates of 100, 9.52, and 1.98 Hz, respectively. The excitation laser power was optimized empirically for each exposure time to achieve sufficient contrast for single-molecule tracking and the powers of the photoswitching laser were also optimized empirically to achieve low enough densities of detections to enable tracking. The excitation laser power was 0.1 mW, 0.6 mW, 2.3 mW and switching laser power was 2.3 µW, 3.9 µW, and 8.5 µW for 500, 100, and 7.5 ms exposures, respectively, as measured at the back focal plane of the excitation objective. The same settings were used to acquire control data at each exposure time on His2B-mEos3.2. For all exposure times, the length of each acquisition was 105 s, corresponding to 200, 1000, and 10,000 frames at 500, 100, and 7.5 ms exposure times, respectively. The acquisition length was set so that sufficient fields of views could be captured in the short interphase times of the early nuclear cycles while also capturing a sufficient number of single-molecule trajectories.

For characterization of single-molecule dynamics at these multiple time scales, both mEos3.2-Bcd and mEos3.2-Zld were measured in a His2B-EGFP background. The His2B-EGFP channel was used to ensure optimal positioning of the sample within the light-sheet, to keep track of progression through a cell cycle, and monitor the development of the embryo. A fortunate bonus was that at 100 ms and 500 ms exposures, there was sufficient excitation of His2B-EGFP from the photoactivation 405 nm laser that we could perform simultaneous imaging of chromatin and single-molecule dynamics (*Figure 1—video 3* and *Figure 1—video 4*).

For quantification of single-molecule mEos3.2-Bcd dynamics in the context of mNeonGreen-Zld, single-molecule data was acquired for 1 s (10 frames at 100 ms exposure times), followed by 10 frames of acquisition in the mNeon channel at 10 ms exposure times, and this sequence was then repeated 100 times. The sum of the 10 mNeonGreen images was then calculated to effectively provide a 100 ms exposure image. This scheme was designed such that the dynamic motion of Zld could be captured in addition to the binding kinetics of Bcd with sufficient temporal resolution without having to modify the LLSM control software. The rest of the imaging parameters were kept identical to those described above. For all single-molecule experiments, nuclei from at least three embryos were measured spanning a range of anteroposterior positions and at nuclear cycles ranging from 12 to 14.

Localization and tracking of single molecules was performed using a MATLAB implementation of the dynamic multiple-target tracing algorithm (*Sergé et al., 2008*) as previously described (*Mir et al., 2017*; *Hansen et al., 2018*; *Hansen et al., 2017*; *Teves et al., 2016*).

## Mean square displacement analysis

Mean Square Displacement curves were calculated using the open source msdanalyzer package (*Tarantino et al., 2014*). For analysis of sub-diffusive motion MSD/τ curves for His2B, Zld, and BCD plotted on log-log-scale. As for anomalous diffusion $MSD(\tau)=\Gamma\tau^{\alpha}$, where $\alpha$ is the confinement factor the $\log(MSD/\tau)=\log(\Gamma) + (\alpha-1)\tau$ (*Izeddin et al., 2014*). The log of the MSD/τ was thus used to estimate the range of $\alpha$ values for each protein.

## Analysis of short exposure (10 ms) single-molecule trajectories

Single-molecule trajectories were analyzed using Spot-on (*Hansen et al., 2018*), a freely available open-source software (https://gitlab.com/tjian-darzacq-lab/spot-on-matlab) based on a model previously introduced in *Mazza et al. (2012)* and modfied in *Hansen et al. (2017)* to exclude state transitions. In brief, Spot-On performs fits to the distribution of displacements at multiple frameshifts to a two-state kinetic model and provides estimates of the fraction of molecules bound and free, and the

corresponding apparent diffusion coefficients for each state (*Figure 3—figure supplement 1*) and corrects for the probability of molecules diffusing out of the axial detection range. We performed fitting using the following parameters: Gaps Allowed: 1, Jumps to Consider: 4, TimePoints: 8, Observation Slice: 0.8 μm, Fit Iterations 5. The fit parameters for each data set are summarized in *Figure 3—figure supplement 1*. Data are represented as the mean over the three embryo replicates ±SEM.

## Calculation of residence times from long exposure single-molecule trajectories

Imaging with sufficiently long exposure times effectively blurs out fast-moving molecules into the background while molecules stably bound for a significant duration of the exposure time are imaged as diffraction limited spots (*Hansen et al., 2017*; *Watanabe and Mitchison, 2002*; *Mir et al., 2017*; *Teves et al., 2016*; *Chen et al., 2014a*; *Chen et al., 2014b*). Thus, the trajectories from the 500 ms datasets are used to infer the genome average long-lived (specific) binding times.

To infer the residence time, the length of trajectories in time is used to calculate a survival probability (SP) curve (1- cumulative distribution function of trajectory lengths). Since the SP curve contains contributions from non-specific interactions, slowly moving molecules, and localization errors a double-exponential function of the form $SP(t)=F*(exp(-k_{ns}*t))+(1 F)(exp(-k_s*t))$ is fit to the SP curve, where $k_{ns}$ is the off-rate for the short-lived (non-specific) interactions and $k_s$ correspond to the off-rate of long lived (specific) interactions (*Chen et al., 2014a*; *Chen et al., 2014b*; *Hansen et al., 2017*; *Mir et al., 2017*) (*Figure 4—figure supplement 2*). For fitting purposes, probabilities below $10^{-3}$ are not used to avoid fitting the data poor tails of the distribution. An objective threshold on the minimum number of frames a trajectory lasts is then used to further filter out tracking errors and slow-diffusing molecules (*Mazza et al., 2012*; *Hansen et al., 2017*). The objective threshold is determined by plotting the inferred slow rate constant and determining where values converge to a single value. Although the 500 ms Bcd data set converges at two frames (1 s), the Zld data set converges at four frames (2 s) (*Figure 4—figure supplement 2B–C*). The survival probability distribution for Zelda is likely dominated by short-lived interactions at shorter timescales and is most likely a reflection of the same complex mixed population (specific and non-specific DNA binding, along with another population whose motion is constrained perhaps by protein-protein interactions) we observed in the MSD curves (*Figure 2B*). Thus, a four-frame threshold was used for the calculation of the specific residence time.

Next, since the inferred $k_s$ as described above is biased by photobleaching, and nuclear and chromatin movement, bias correction is performed using the His2B data as $k_{s,true}=k_s-k_{bias}$, where $k_{bias}$ is the slower rate from the double-exponent fit to the His2B SP curve as described previously (*Teves et al., 2016*; *Hansen et al., 2017*; *Chen et al., 2014a*; *Chen et al., 2014b*). This correction is based on the assumption that photobleaching, unbinding, and loss of trajectories from motion are all independent Poisson processes. The genome wide specific residence time is then calculated as $1/k_{s,true}$. The effectiveness of this bias correction is checked by calculating the residence time from both the 100 ms and 500 ms frame rate data and observing convergence to within 1 s (*Figure 4—figure supplement 2C*).

## Fluorescence recovery after photobleaching

FRAP was performed on a Zeiss (Germany) LSM 800 scanning confocal microscope equipped with several laser lines, of which the 488 nm laser was used for all experiments described here. Images were collected using a Plan-Apochromat 63 × 1.40 NA oil-immersion objective using a window 50.7 μm by 3.6 μm. Bleaching was controlled by the Zen software, and experiments consisted of 10 frames collected before the bleach and 1000 frames collected after at a frame rate of 24 ms. In each frame, five circular bleach spots of 1 μm diameter were chosen to be a sufficient distance from nuclear edges. The spots were bleached using maximum laser intensity, with dwell time adjusted to 0.57 μs, which was chosen because it gave a sufficiently deep bleach of Bicoid, the fastest-recovering molecule we studied. Total bleach time was 1.5 s.

We collected data from at least three embryos for each molecule studied. Nuclei in the early embryo are highly mobile, and we found that the most reliable method to find stable nuclei was to simply collect many movies and select the ones in which nuclei remain stable for the duration of the

experiment. We collected movies with stable nuclei for a total of at least 50 bleach spots (50 nuclei) total for each molecule. To quantify and bleach-correct FRAP data, we used a custom-written MAT-LAB software pipeline (*Mir and Stadler, 2018*; copy archived at https://github.com/elifesciences-publications/MirStadler_2018). Briefly, for each frame we manually select several 'dark' spots that are not within nuclei and several 'control' spots that are within bleached nuclei but well-separated from the bleach spot. We use a 600 nm diameter circle to calculate the signal at the spots in order to make the measurement robust to small chromatin movements. For each frame, the mean of the dark spots was subtracted from the bleach spot values (background subtraction), and individual bleach spot values were divided by the mean of the control spots to correct for the reduction in total nuclear fluorescence. Finally, the values for each spot were normalized to its mean value for the ten pre-bleach frames. We observed that chromatin movement occasionally causes the bleach spot to drift far enough to affect the signal, so we manually curated resulting correct traces to remove anomalous spots. This culling resulted in 27–40 quality recovery curves for each molecule. These curves were averaged for each molecule, and the mean recovery curve was used in figures and fitting.

We fit resulting FRAP curves to the reaction-dominant model (*Sprague et al., 2004*):

$$FRAP(t) = 1 - Ae^{-k^{at}} - Be^{-kbt}$$

From these fits, we used the slower coefficient to estimate the time to half-recovery for the population of bound molecules.

## Analysis of single-molecule binding in the context of Zld density

To analyze single-molecule trajectories of Bcd and Zld in the context of Zld density first, a relative density map for each nucleus was calculated. For each reconstructed 100 ms exposure Zld image, first each nucleus was identified and segmented out of the image using an in house segmentation algorithm built in MATLAB (*Figure 6—figure supplement 1*). First, the grayscale image was Gaussian filtered with a sigma of 5 pixels to enhance the contrast of the nuclei, the filtered image was then thresholded using the inbuilt adaptive threshold function in MATLAB with a sensitivity value set to 0.6. A morphological dilation was then performed on the binary mask using a disk structuring element with a radius of 3 pixels and multiplied with hand drawn mask to remove edges of the embryo and non-cortical regions deep where no nuclei were present or imaging contrast was low. Holes within the nuclei binary mask were then filled using the MATLAB imfill function. A label matrix was generated from the resulting binary mask and the size distributions and eccentricities of segmented regions were calculated, an area and eccentricity cutoff was then applied to remove false positives to generate the final label matrix. Label matrices were then further curated to remove false positives. A relative density map was calculated for each nucleus individually by assigning each pixel in the nuclear value the percentile range it fell in over the entire distribution of intensity values in the nuclear area in with a resolution of 1 percentile. Each single-molecule trajectory was then assigned a relative density value based on the mean density of the pixels it fell in during the course of the trajectory. From visual examination, we determined that a 85 relative density value threshold was reliable in differentiating the highest enriched Zld regions, corresponding to hubs, from the rest of the nucleus. Fold change in densities of detections were calculated by counting the total number of trajectories in areas of relative density greater than 85 vs. the rest of the nucleoplasm. As the single-molecule trajectories from this data set are limited in length to 1 s, an accurate estimate of the residence time from fits to the survival probability distribution could not be obtained as was done above.

## Analysis of protein distribution in context of transcription dynamics

Two-color 4D LLSM imaging was performed on embryos with the MS2-tagged *hb* BAC (*Figure 7—figure supplement 1*) and expresing MCP-mCherry crossed with either mNeonGreen-Zld or eGFP-Bcd embryos. Z-stacks of 61 slices were acquired with a spacing of 250 nanometers to cover a range of 15 µm with an exposure time of 80–100 ms in each channel at each slice. Images in both channels were acquired at each z-position sequentially before moving to the next slice. The time between each volume acquired was ~5 s, the total length of the acquisition varied but at least one complete nuclear cycle was imaged for each embryo. The field of view for each embryo was centered at between 25% and 35% of the embryonic length from the anterior tip of the embryo to ensure that

all nuclei in the image were within the hunchback expression domain. Data from a total of six embryos each for mNeonGreen-Zld or eGFP-Bcd were analyzed.

To analyze the distribution of Zld or Bcd around sites of active *hb* transcription the signal from the MS2 site was used as a marker for the active locus. Each MS2 site was localized through a custom built detection software (*Figure 7—figure supplement 1*). First, the data was manually examined and annotated to simplify the segmentation procedure by only considering frames in which transcription was occuring. A 3D difference of gaussian image was then calculated at each frame to enhance the contrast of the MS2 site, a global threshold was then applied to generate a 3D binary mask for each frame. The binary mask was filtered to remove structures too big or too small to be from a MS2 site and a label matrix was generated. The xyz weighted center of each labeled region was then used to calculate line profiles extended one micron from the center of the region in each direction. The ratio of the maximum and minimum values in the line profile were used to determine if the labeled region was in a nucleus. This calculation is effective as in the MCP-mCherry channel the nucleoplasm around the MS2 site appears dark whereas in the remainder of the embryo the background is high, thus labeled regions with low-contrast ratios were discarded. The maximum value of the profile of the remaining labeled regions was then used to localize the center of the active locus. The detected loci in each frame were then connected in time using a nearest neighbor algorithm.

The position list of the detected and tracked active loci were then used to crop a 2.18 μm window around the center of each locus in x-y, if any part of the window did not lie within a nucleus the locus was not considered for further analysis. For the remaining loci, a control window was cropped at a distance of 2.6 μm from the center of the locus in x-y. If a control window could not be found that did not completely lie within the nucleus the corresponding locus was also not considered for further analysis. In this manner a total of 3943 and 6307 windows centered around loci and corresponding control points were accumulated from the Bcd and Zld datasets, respectively. The mean image at the locus was then calculated (*Figure 7C*, *Figure 7—video 4*, *Figure 7—video 5*) and a radial profile was calculated for Zld or Bcd centered at the active locus or the random control site. The radial profiles were then normalized to one at the maximum radius.

## Acknowledgements

We thank Robert Tjian for useful discussion and extensive advice on the science and the manuscript. We thank Jacques Bothma, Simón Álamos and Hernan Garcia for useful discussions and providing the *hb-MS2* and MCP-mCherry fly lines. We also acknowledge the advice and comments provided by all members of the Eisen, Darzacq, Tjian, and Garcia labs over the course of this work. This work was supported by a Howard Hughes Medical Institute investigator award to MBE; XD acknowledges support from California Institute of Regenerative Medicine (CIRM) grant LA1-08013 and the National Institutes of Health (NIH) grants UO1-EB021236 and U54-DK107980; MRS was supported by an American Cancer Society postdoctoral fellowship; MMH was supported by NIH GM111694.

## Additional information

### Funding

| Funder | Grant reference number | Author |
| --- | --- | --- |
| American Cancer Society | | Michael R Stadler |
| National Institutes of Health | GM111694 | Melissa M Harrison |
| National Institutes of Health | UO1-EB021236 | Xavier Darzacq |
| California Institute for Regenerative Medicine | LA1-08013 | Xavier Darzacq |
| National Institutes of Health | U54-DK107980 | Xavier Darzacq |
| Howard Hughes Medical Institute | | Michael B Eisen |

The funders had no role in study design, data collection and interpretation, or the decision to submit the work for publication.

### Author contributions
Mustafa Mir, Conceptualization, Resources, Data curation, Software, Formal analysis, Supervision, Validation, Investigation, Visualization, Methodology, Writing—original draft, Project administration, Writing—review and editing; Michael R Stadler, Conceptualization, Data curation, Software, Formal analysis, Validation, Investigation, Visualization, Methodology, Writing—original draft, Writing—review and editing; Stephan A Ortiz, Data curation, Software, Formal analysis; Colleen E Hannon, Validation, Writing—review and editing; Melissa M Harrison, Resources, Funding acquisition, Writing—review and editing; Xavier Darzacq, Conceptualization, Resources, Supervision, Funding acquisition, Investigation, Writing—original draft, Project administration, Writing—review and editing; Michael B Eisen, Conceptualization, Supervision, Funding acquisition, Investigation, Writing—original draft, Project administration, Writing—review and editing

### Author ORCIDs
Mustafa Mir (ID) http://orcid.org/0000-0001-8280-2821
Michael R Stadler (ID) http://orcid.org/0000-0002-3333-4184
Colleen E Hannon (ID) http://orcid.org/0000-0002-4402-8107
Melissa M Harrison (ID) http://orcid.org/0000-0002-8228-6836
Xavier Darzacq (ID) http://orcid.org/0000-0003-2537-8395
Michael B Eisen (ID) http://orcid.org/0000-0002-7528-738X

### Decision letter and Author response
Decision letter https://doi.org/10.7554/eLife.40497.043
Author response https://doi.org/10.7554/eLife.40497.044

## Additional files

### Supplementary files
• Transparent reporting form
DOI: https://doi.org/10.7554/eLife.40497.038

### Data availability
All data generated or analysed during this study are included in the manuscript and supporting files. Source videos are also available on http://eisenlab.org/hubs and software through GitHub (https://github.com/meisenlab/MirStadler_2018; copy archived at https://github.com/elifesciences-publications/MirStadler_2018).

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
