## [Decision Letter]

[**Editorial note:** This article has been through an editorial process in which the authors decide how to respond to the issues raised during peer review. The Reviewing Editor's assessment is that all the issues have been addressed.]

Thank you for submitting your article "Dynamic multifactor hubs interact transiently with sites of active transcription in *Drosophila* embryos" for consideration by *eLife*. Your article has been reviewed by two peer reviewers, and the evaluation has been overseen by a Reviewing Editor and Kevin Struhl as the Senior Editor. The following individual involved in review of your submission has agreed to reveal his identity: Justin Crocker (Reviewer #1). Reviewer #2 remains anonymous.

The Reviewing Editor has highlighted the concerns that require revision and/or responses, and we have included the separate reviews below for your consideration. If you have any questions, please do not hesitate to contact us.

The significance of the work is that it adds to the evolving knowledge that transcription factors can form concentrations around active genes. Both reviewers recognize this but have some suggestions for improvements in the manuscript clarity, the need for controls and the data analysis, particularly justification of the modeling and quantification of the dynamics, including FRAP.

Separate reviews (please respond to each point):

*Reviewer #1:*

In the manuscript "Dynamic multifactor hubs interact transiently with sites of active transcription in *Drosophila* embryos," Mir and colleagues explore the nuclear organization and interactions of the transcription factors Zelda and Bicoid. They find that both factors interact with DNA with high off-rates, forming dynamic sub nuclear hubs. Finally, they demonstrate that the TF hubs are short-lived at sites of active transcription. Together, this is an excellent manuscript that contributes to a growing body of evidence that transcription factor interactions with regulatory regions are highly dynamic and can occur in local hubs of high concentration.

That said, I have recommendations for the others:

First, while the authors outline the tagging of endogenous proteins with mEos3.2 etc., the characterization and presentation were cursory. The authors should outline what they did to test for possible aggregation when these FPs are fused to their target proteins. They could be altering intrinsic clustering or polymerization of the native proteins. They should verify the spatial organization derived from FPs with alternative approaches, such as immunohistochemistry or by using SNAP/ HALO.

For a general audience, an explanation of how the authors controlled for bleaching rates and photophysical properties of the tagged proteins would clarify the manuscript.

In all bar graphs throughout the paper, the authors should overlay the data points, show violin plots (or a similar variant), or both. It would be great to see the distribution of the data particularly if they are not single exponential.

Did the authors test any other kinetic models? The authors need to justify why they chose the two-state model-it would be great to see how a 3-state kinetic model fits the data, including fast or slow free molecules, linear diffusion, etc.

In Figure 6, what did the enrichment look across the embryos? It would aid the reader to show nuclei, as done in 6C, for regions across the embryo. It is also highly unclear what Anterior, middle, posterior, Zld refer to. Additionally, the arrow in Figure 6C is hard to see. Do the authors have videos stills they could include in the main figure of Bcd molecules moving into Zld hubs and changing dynamics?

For Figure 7, and the associated data, do the spatiotemporal distributions change across the embryo (i.e., anterior-to-posterior)?

This manuscript should be published provided the authors address these details, which could be done mostly through their existing data with minimal new experiments.

*Reviewer #2:*

In this work Mir et al. use light-sheet microscopy combined with photoactivation, single-molecule tracking and high speed 4D imaging to study nuclear organization, dynamics and potential interactions of the Bicoid and Zelda transcription factors in developing fruit fly embryos. For both factors, they detect several populations of molecules with different motilities and show that their distribution within the nucleus is not homogenous with structures of higher concentration (hubs). These hubs have short half-lives and partially colocalize with each other and with sites of active Bicoid transcription. In light of these observations, they propose that the interaction between the DNA bound transcription factors with the transcription machinery is much more dynamic than anticipated to allow for diverse activities at target sites.

Although there is a clear potential in the approaches developed, which combined advanced genome editing with a cutting edge imaging expertise, I found that the analysis of the data remains superficial and only partially support the conclusion. I also think that at many instances, there is a need for clarification to reach the large audience expected for *eLife*. I have listed below the major points that should be clarified before considering publication.

1) Concerning the Cas9-mediated insertion of highly fluorescent tags of endogenous, several linkers and tags were used looking for their viable insertions at specific positions. It is not clear which ATG was used to initiate translation. This is at least particularly critical in the case of Bicoid whose translation is known to be triggered by egg laying through a complex process which might be disrupted by just adding an upstream initiation codon. I found the test for viability as a guaranty a little bit "dangerous": it is possible that internal initiation could allow expression of a sufficient amount of untagged protein for viability (in this case viability would not be a strong argument for the proper expression of the fusion protein). Concerning the approach as it is also novel, it would be useful to have a table indicating the number of lines checked by PCR with the different features and how many turned out to be eliminated. Finally, it is surprising that the authors did not try several sites of insertion. They assume that inserting the fluorescent tag at the N-terminus of Bicoid or Zelda should not affect their function (which is an assumption that could biased the downstream screen). Same is true for the fluorescent tag at the C-terminus of H2B. The data would be much stronger if reproduced for another site of insertion. Another important control is the behavior of the fluorescent tag alone and/or some experiments with mutated protein (for instance by destructing DNA binding) showing different behaviors.

2) The authors have to be careful with the use of H2B as a control. First, there is likely a population of free H2B in the nucleus and clearly imaging H2B-eGFP does not exclusively enable to image chromatin (this indication in the Figure 1 legend is misleading). Also, H2B is bound at many more sites within the genome than Bicoid or Zelda. It is thus dangerous to use the slow mobility of H2B as a standard to identify the populations of Bicoid and Zelda molecules bound to DNA. It is also possible that the proposed transient interaction of Bicoid and Zelda to DNA correspond to non-specific binding and that specific binding would be much more rare events (given the number of expected binding sites) and would not be detected in the assay. Since these transient interactions are strongly emphasized by the authors as an argument for the dynamic behavior of the transcription factors, this point has to be clarified. Also, could-it be that slow motion (what the author name "bound") is 1D diffusion? In general, in the text "bound/unbound" should be replaced by "slow/fast moving".

3) Concerning the FRAP experiment, the recovery plot should start at 0 and does not. This is in general an indication that some of the molecules are moving too rapidly to be properly analyzed by FRAP (Waharte et al., BioPhys J, 2005). The plot shown in Figure 2—figure supplement 1 is not very satisfying because the lowest point of the Bicoid recovery is unknown (between 0.2 and 0.6). It is also not clear why the FRAP experiment validate the use of H2B a suitable control. The authors should be more explicit. In addition, fluorescent correlation spectroscopy data, which is used also to identify various population of highly mobile/less mobile molecules, are already available for H2B and Bicoid. The data presented here and their conclusion concerning motilities (MSD) have to be discussed in light of these already published information.

4) The fast dynamics of interphase hubs is a strong argument used by the authors in the overall message they want to deliver. This cannot be only based on qualitative arguments and the quantitative description of this dynamics has to be provided. In particular, the data presented Figure 5 and discussed in paragraph two of subsection “Zelda and Bcd form dynamic subnuclear hubs” have to be quantified.

5) Concerning the enrichment of Bcd and Zld at the hb transcription sites, the 1.3 versus 1.1 relative enrichment observed for Bcd seems to be due to lower background (Figure 7C). It would be important to comment on this and provide the data before normalization in the form of line profiles (one radius for X axis and intensity for the Y axis). This would help understand the data, get an idea of the noise in the measurements and determine whether the 1.1 to 1.3 increase could be meaningful. Also, it is not clear why the enrichment of Bcd binding events in Zld hubs could not be due to an increase k_off_. This needs to be clearly explained and rationalized.

Minor Comments:

1) Introduction paragraph nine: beginning of the sentence is unclear given the cited reference.

2) Figure 1B: yellow arrows are very difficult to distinguish from white arrows.

3) Subsection “Zelda hubs are not stably associated with sites of active transcription” first paragraph: be more explicit on the number of sites given also the fact that Figure 7—figure supplement 1) does not at all provide any information related.

4) Discussion section: transient binding (if this is really what is detected in this study) is not inconsistent with ChIP experiments showing increase occupancy of Bcd in Zld binding regions which are performed on fixed material and provide only a "cumulative" snap-shots (time scale of the fixation process) of the state of the bound regions. It might just be a matter of time-scale.

5) Discussion, paragraph five: not clear to what "previous reports" the authors refers to. Provide references.

6) Discussion, paragraph five: better explain what is meant by "time-average accumulation" (and also the reference to the work on Bcd at hb).

7) Discussion section: there is also a possibility that the hubs reflect some kind of structure formed when the transcription factors are inactive (the authors mention that they are "more pronounced at times when no transcription happens") and that the real interaction events responsible for activity are not detected because they are too rare. For instance: if the activation event requires 6 Bcd proteins bound at a given site, what is the probability that this event would be detected with photoactivable approach unless the interaction is long enough (what is the time-scale that would allow to capture these events).

8) Materials and methods, paragraph four: indicate the promoter allowing expression of H2B.

---

## [Author Response]

Reviewer #1:

In the manuscript "Dynamic multifactor hubs interact transiently with sites of active transcription in Drosophila embryos," Mir and colleagues explore the nuclear organization and interactions of the transcription factors Zelda and Bicoid. They find that both factors interact with DNA with high off-rates, forming dynamic sub nuclear hubs. Finally, they demonstrate that the TF hubs are short-lived at sites of active transcription. Together, this is an excellent manuscript that contributes to a growing body of evidence that transcription factor interactions with regulatory regions are highly dynamic and can occur in local hubs of high concentration.

We thank Dr. Crocker for his positive comments and recognition of the manuscript’s contribution to the understanding of transcriptional regulation.

That said, I have recommendations for the others:First, while the authors outline the tagging of endogenous proteins with mEos3.2 etc., the characterization and presentation were cursory. The authors should outline what they did to test for possible aggregation when these FPs are fused to their target proteins. They could be altering intrinsic clustering or polymerization of the native proteins. They should verify the spatial organization derived from FPs with alternative approaches, such as immunohistochemistry or by using SNAP/ HALO.

We have observed similar spatial distributions of Zelda protein via live imaging with a variety of tags, including sfGFP, mCherry, mEos3.2, and mNeonGreen. Furthermore, the existence of hubs has been shown independently using IF, GFP, and mCherry by Dufort et al. (BioRxiv, 2018), as well as by IF experiments conducted in our group. The consistency of these observations, together with the well-characterized monomeric nature of mEos3.2, mNeonGreen, and mCherry make it extremely unlikely that the aggregation and clustering we are observing is a result of the tags themselves, and strongly suggests that the phenomena we describe reflect the genuine in vivo behavior of these proteins. With regards to using SNAP/HALO, although we have generated lines with these tags, we have not been able to reliably deliver the dyes to the early embryo in a manner that supports continued development. The early embryo is uniquely sensitive to both permeabilization and microinjection, and we have thus decided to avoid using organic dyes until we can deliver these dyes in a manner that is less perturbative.

For a general audience, an explanation of how the authors controlled for bleaching rates and photophysical properties of the tagged proteins would clarify the manuscript.

Photobleaching correction is only applied to calculate the off-rates of DNA binding in the 500 and 100 ms data using the corresponding H2B measurements. In the original manuscript the details of this correction were explained in the Materials and methods section under “Calculation of residence times from long exposure single molecule trajectories” and Figure 4—figure supplement 1. To further clarify the basis of this correction we have now added the following to the Results section under “Bicoid and Zelda bind transiently and have large free populations”

“This correction is based on the fact that the long-lived population of His2B is associated with chromatin much longer than the dynamic range of our measurement and that the maximum trajectory lengths that we measure for His2B are thus only limited by photo-bleaching and defocalization (see Materials and methods for more details) (Hansen et al., 2017).”

In all bar graphs throughout the paper, the authors should overlay the data points, show violin plots (or a similar variant), or both. It would be great to see the distribution of the data particularly if they are not single exponential.

Wherever we use bar graphs they are used to display results from model fits to multi-population distributions. These bar graphs show the means of 3 biological replicates. The full distributions of the data used for fitting are always shown either next to the bar plots (Figure 3A for kinetics, Figure 4A-B for residence times) as well as in supplementary figures.

Did the authors test any other kinetic models? The authors need to justify why they chose the two-state model-it would be great to see how a 3-state kinetic model fits the data, including fast or slow free molecules, linear diffusion, etc.

We agree with the reviewer that it is possible, and in fact likely, that the populations we call bound and free are composed of more complex sub-states including slowly-diffusing molecules or linear diffusion. We did experiment with 3 state models, and this model did not sufficiently improve the fit to justify the use of additional parameters. For all our data, a 2 state model is sufficient to fit the distributions.

We thank the reviewer for pointing out the need to justify and clarify this point and we have now including the following in the revised manuscript under the Results subsection “Bicoid and Zelda bind transiently and have large free populations”:

While it is likely that the identified immobile and free diffusing populations contain more complex sub-populations, for example molecules exhibiting 1-D diffusion on DNA, a 2-state model accurately fit the displacement distributions for all 3 proteins and using a higher number of states did not significantly improve the model fit to justify them.

We have also modified other instances in the text and Figure 3 to clarify the distinction between bound and immobile when discussing the high frame-rate data. However in the case of the 100 ms and 500 ms data, we still refer to the detected molecules as bound, as already justified extensively in the text.

In Figure 6, what did the enrichment look across the embryos? It would aid the reader to show nuclei, as done in 6C, for regions across the embryo.

What we refer to as “enrichment” in Figure 6C is regions with high Zelda density. What we report on here is not the change in the enrichment of Zelda across the embryo, which we observe is not dependent on position along the A-P axis but rather the enrichment of Bicoid binding events in Zelda high regions.

We recognize that the way we used the word enrichment in this context is confusing as we are presenting data on both the local enrichment of Zld in terms of its spatial distribution and the enrichment of single molecule binding events of Bicoid within regions of high Zld density (with these enrichments split by anteroposterior position). We regrettably used the words “high density” and “enrichment” interchangeably in this section and corresponding figures, and we have amended this.

We now use the word enrichment to consistently and only refer to the enrichment of Bcd binding events within Zld hubs and refer to Zld levels as Zld “density” and have made the following changes:

In the Results subsection “Bicoid binding events are enriched in Zelda hubs” all instances where we are referring to the spatial distribution of Zld as “enrichment” have been replaced with “density” whereas the enrichment of Bcd binding events in areas of high Zld density are now solely referred to as “enriched”. We have also provided an additional clarification that high density Zelda regions correspond to hubs.

“Using the bulk Zld data we partitioned nuclei into regions of high and low relative Zld density (Figure 6B-C and Figure 6—figure supplement 1), where high density regions correspond to Zld hubs.”

Figure 6C: The label for the color bar has been changed to “Relative Zelda density”

Figure 6C caption has been changed: Relative Zelda Density map for the nucleus shown in (B), the arrow on the colorbar indicates the threshold for defining a region as high density (corresponding to hubs).

Figure 6D: Y-axis label has been changed to: “Fold Enrichment of Binding Events/µm^2^) in Zld Hubs.

Figure 6—figure supplement 1: Title has been changed to “Analysis of Zelda density” and caption and text in the figure has been modified to replace all instances of the word “enrichment” with “density”.

Figure 6—figure supplement 1D: labels on bars have been modified to “High” and “Low” from “Enriched” and “Not Enriched”.

Figure 6—figure supplement 2: all instances of “enrichment” have been replaced with “density” or “zelda density” in both the figure and captions.

The title of Materials and methods subsection “Analysis of single molecule binding in the context of Zld enrichment” has been changed to “Analysis of single molecule binding in the context of Zld density” and all usage of the word “enrichment” within the section has been changed to “density.

To further clarify that the spatial distribution of Zld remains unchanged across the anteroposterior axis we add the following to the corresponding Results subsection: “The enrichment of Bcd trajectories within the high density Zld regions increases along the anteroposterior axis of the embryo as the Bcd concentration decreases to an excess of around four-fold in the posterior (Figure 6D) whereas the Zld spatial distribution remains unchanged”.

It is also highly unclear what Anterior, middle, posterior, Zld refer to.

We have modified the x-axis label on Figure 6D to read Bcd in Anterior, Bcd in Middle, Bcd in Posterior and also the y-axis label as described above for clarity. We have also edited the caption to read: “Fold change in enrichment of single molecule trajectories of Bcd in the anterior, middle, and posterior positions, and of Zld in Zld hubs vs. the rest of the nucleoplasm.”

Additionally, the arrow in Figure 6C is hard to see.

The arrow has been repositioned and enlarged to make it easier to see.

Do the authors have videos stills they could include in the main figure of Bcd molecules moving into Zld hubs and changing dynamics?

It would be fantastic if it was possible to have such data with our current system however as explained in the main text:

“To strike a balance between the constraints of the imaging system, the dynamic range of the single molecule trajectories, and the fast dynamics of Zld hubs (Figure 5 and Figure 6A) we acquired a bulk fluorescence image of mNeonGreen-Zld with a 1 sec acquisition time followed by 10 frames of single molecule images with a frame rate of 100 ms (Videos 14)”.

Given the constraints of imaging, this experiment was designed to be sensitive to the bound population of Bcd (based on our measurements of its kinetics) and in this case we thus only detect molecules of Bcd that are immobile for at least 100 ms and only track them for a total of 1 second.

For Figure 7, and the associated data, do the spatiotemporal distributions change across the embryo (i.e., anterior-to-posterior)?

In this experiment all data was acquired in a narrow region of the hunchback anterior expression domain in nuclear cycle 13 and early nuclear cycle 14 and is thus not designed to look at changes in the spatiotemporal distributions across the embryo. We agree that this is an important question given our findings of a stronger enrichment of Bcd binding within Zld hubs at more posterior positions, however answering this question requires utilizing a Bcd target gene tagged with MS2 that is expressed at more posterior positions. We plan to systematically study this question in our future work.

This manuscript should be published provided the authors address these details, which could be done mostly through their existing data with minimal new experiments.

We thank Dr. Crocker for his comments, which have greatly improved the clarity of this manuscript, and hope that it is now more accessible to readers.

Reviewer #2:

[…] Although there is a clear potential in the approaches developed, which combined advanced genome editing with a cutting edge imaging expertise, I found that the analysis of the data remains superficial and only partially support the conclusion. I also think that at many instances, there is a need for clarification to reach the large audience expected for eLife. I have listed below the major points that should be clarified before considering publication.

We thank the reviewer for their careful reading and constructive comments. We address each of the concerns raised by the reviewer fully in the revised manuscript as outlined below.

1) Concerning the Cas9-mediated insertion of highly fluorescent tags of endogenous, several linkers and tags were used looking for their viable insertions at specific positions. It is not clear which ATG was used to initiate translation. This is at least particularly critical in the case of Bicoid whose translation is known to be triggered by egg laying through a complex process which might be disrupted by just adding an upstream initiation codon. I found the test for viability as a guaranty a little bit "dangerous": it is possible that internal initiation could allow expression of a sufficient amount of untagged protein for viability (in this case viability would not be a strong argument for the proper expression of the fusion protein).

The reviewer raises significant points. First, it is important to clarify the sites of insertion, specifically which ATG was used. We have amended the manuscript as follows:

“Fluorescent tags were inserted at ATG located at 3R:6759268 for Bicoid and at X:19782283 for Zelda (dm6 coordinates). For both Zelda and Bicoid, the natural start ATG was removed, and the fluorescent protein and linker were fused with the first amino acid after the initiation Methionine residue.”

The use of organismal viability as a test for the normal function of the fusion proteins relies on the assumption that, in homozygous animals, the fusion protein is the only version produced from the target gene. As the reviewer points out, production of un-tagged protein, via internal initiation, cryptic splicing, or proteolytic cleavage, would violate this assumption, leaving the possibility that fusion proteins are non-functional and viability is accomplished instead by the pool of un-tagged protein. To address this possibility, we performed Western blots on 2 hour embryos for all of the newly-generated fusion protein lines referenced in the paper (Figure 1—figure supplement 1). For Zelda, we detect no protein at the size expected for un-tagged protein in the mEos3.2, mNeonGreen, or mCherry lines. For mEos3.2-Bicoid, we observe a strong band at the expected size of the fusion protein, and a very faint smear in the area of the un-tagged protein. This smear likely represents background staining, which is obscured by the strong band for native protein in wild-type samples, but in any case represents a level of protein too low to account for the viability of these lines, even if it does represent Bicoid degradation products (Liu et al., 2013 PNAS, Hannon et al., 2017. Both authors were unable to generate viable lines expressing less than ~0.4x Bicoid protein).

Methods: For each genotype indicated, 50-60 embryos aged 2 hours at 25º were dechorionated in bleach, rinsed in salt solution (NaCl with TritonX-100) and flash frozen. Frozen embryos were homogenized in 50 µL of sample buffer and 20 µL of sample per lane was loaded onto two separate 4-15% SDS-PAGE gels (Bio-Rad Cat # 4561083DC). Western blots were performed using rabbit polyclonal α-Bcd or α-Zld primary antibodies and a goat α-rabbit HRP conjugated secondary antibody (Thermo Fisher Cat # 31460).

Concerning the approach as it is also novel, it would be useful to have a table indicating the number of lines checked by PCR with the different features and how many turned out to be eliminated.

This is a good suggestion, and would indeed be useful as a guide to others who might wish to use similar approaches. We have included representative statistics for a pooled Zelda transgenesis experiment in the Materials and methods section:

“Typically, we find that ~10-40% of F1 animals contain insertions. For example, for a pooled injection of three Zelda tagging constructs, we screened 79 F1 animals and recovered 36 hits, of which we screened 19 to determine the identify of the inserted tag, recovering 8 GFP-Zld, 6 mEos3.2-Zld, and 5 HaloTag-Zld. We find that efficiency varies for different genes and injections, but these results are fairly typical.”

We have now performed similar experiments for dozens of target genes, and will publish the complete data for these experiments elsewhere.

Finally, it is surprising that the authors did not try several sites of insertion. They assume that inserting the fluorescent tag at the N-terminus of Bicoid or Zelda should not affect their function (which is an assumption that could biased the downstream screen). Same is true for the fluorescent tag at the C-terminus of H2B. The data would be much stronger if reproduced for another site of insertion. Another important control is the behavior of the fluorescent tag alone and/or some experiments with mutated protein (for instance by destructing DNA binding) showing different behaviors.

The reviewer raises a significant point related to any study of this kind: the addition of a fluorescent tag necessarily alters the behavior of the fused protein. For the proteins we studied, we were fortunate in that previous studies had examined fusion products of each. For Bicoid, N-terminal fusions are well-characterized and have been shown to phenocopy wild-type protein (e.g., Gregor et al., 2007 or Hannon et al., 2016). Likewise, our C-terminal fusions of H2B were based on constructs used successfully in mammalian cell culture experiments (Hansen et al., 2018). Finally, N-terminal Zelda fusions were previously shown to be well-tolerated (Hamm et al., 2017). We thus began our tagging experiments using the insertion sites previously shown to work, and independently validated each tag-linker-protein combination. The combination of previously characterized insertion sites, phenotypic viability of homozygotes, and confirmation that viability is effected exclusively by fusion proteins (see response to reviewer 2, comment 2) gives us high confidence that our fusion proteins are functional and serve as reasonable approximations of the behavior of un-tagged wild-type protein.

Nonetheless we agree with the reviewer that using multiple, distinct approaches to generate fusion proteins and the generation of additional control lines would be desirable, and these are the subject of ongoing work in our labs.

2) The authors have to be careful with the use of H2B as a control. First, there is likely a population of free H2B in the nucleus and clearly imaging H2B-eGFP does not exclusively enable to image chromatin (this indication in the Figure 1 legend is misleading).

Our analysis on the kinetics of H2B single molecule data at high frame rates using Spot-On indicates that ~90% of detected H2B molecules are bound. This fact coupled with our FRAP measurements suggest to us that using H2B as a proxy for directly imaging chromatin is suitable, as is commonly done in the single-molecule field.

Also, H2B is bound at many more sites within the genome than Bicoid or Zelda. It is thus dangerous to use the slow mobility of H2B as a standard to identify the populations of Bicoid and Zelda molecules bound to DNA.

The reviewer is correct in their statement that H2B is bound to many other sites in the genome than Bcd or Zelda, however they misunderstand the use of H2B as a “standard”. The H2B data is not used in any way to identify the bound populations of Bcd and Zld. Rather, independent analysis of the single molecule kinetics at high frame rates and analysis using Spot-On is used to quantify the bound populations of all 3 proteins. We have clarified the use of the H2B data to correct for the effects of photobleaching in the main text (see response to reviewer 1 comment 3 above). This method of estimating specific binding times and bound populations has previously been described and validated extensively in the literature cited in our manuscript.

It is also possible that the proposed transient interaction of Bicoid and Zelda to DNA correspond to non-specific binding and that specific binding would be much more rare events (given the number of expected binding sites) and would not be detected in the assay. Since these transient interactions are strongly emphasized by the authors as an argument for the dynamic behavior of the transcription factors, this point has to be clarified. Also, could-it be that slow motion (what the author name "bound") is 1D diffusion? In general, in the text "bound/unbound" should be replaced by "slow/fast moving".

What we refer to as specific binding events in the manuscript are in fact extremely rare compared to the observed non-specific binding events as is evident in the survival probability distributions shown in Figure 4A and B. The binding times we report as specific are in congruence with measurements on other transcription factors and the observation in the single molecule field that transcription factor binding to DNA is in general transient, a fact with which the field at large is reckoning. We believe that the mystery of how such transient interactions can lead to robust regulation is partly resolved by our observation of hub formation. We discuss this point extensively in the Discussion section of the manuscript.

The reviewer is correct that it is possible and likely that what we refer to as bound and mobile populations when analyzing the fast frame data are composed of more complex sub-populations. We have modified the main text and figures in this regard in all relevant places in the manuscript (see response to reviewer 1 comment 5 above).

3) Concerning the FRAP experiment, the recovery plot should start at 0 and does not. This is in general an indication that some of the molecules are moving too rapidly to be properly analyzed by FRAP (Waharte et al., BioPhys J, 2005). The plot shown in Figure 2—figure supplement 1 is not very satisfying because the lowest point of the Bicoid recovery is unknown (between 0.2 and 0.6). It is also not clear why the FRAP experiment validate the use of H2B a suitable control. The authors should be more explicit.

A review of the FRAP literature will reveal that in general, unless artificially manipulated or normalized to do so, recovery plots do not start at 0. We believe that presenting the data using the normalization we implement is clearer and allows the reader to see (as the reviewer has pointed out) that there is an extremely fast moving population that cannot be bleached effectively.

In our case, the purpose of the FRAP experiment was not to characterize the fast moving population, which is notoriously error prone (see Mazza et al., 2012, and Mueller et al., Curr Opin Cell Biol, 2010) but rather to provide an orthogonal estimate of the residence times of bound Bcd, Zld, and H2B to compare with our measurements from SPT. This is why we use a simple two-exponent reaction dominant model which ignores diffusion (see Hansen et al., 2017 for further discussion on this topic). For these reasons, we use high frame-rate SPT data rather than FRAP to estimate the fast kinetics of each protein, which is a more direct measurement and less model dependent.

Our reaction-dominant modelling of FRAP recoveries is consistent with the residence times we measure with SPT and also reveals that the majority of H2B is bound stably for long timescales, verifying its suitability as a control to correct for the effects of photobleaching and defocalization when estimating residence times from our long exposure time SPT-data. We thank the reviewer for pointing out that this was not clear in the manuscript, and we have now added the following to the text to be more explicit:

“The FRAP data verifies that the majority of His2B molecules are stably bound for significantly longer times than Bcd or Zld and indicates that the single molecule trajectory lengths of His2B are limited by photobleaching and defocalization rather than unbinding.”

In addition, fluorescent correlation spectroscopy data, which is used also to identify various population of highly mobile/less mobile molecules, are already available for H2B and Bicoid. The data presented here and their conclusion concerning motilities (MSD) have to be discussed in light of these already published information.

To our knowledge there is only 1 paper (Porcher et al., Development 2010) that performs FCS on Bicoid within the nuclei of the *Drosophila* embryo. The data from this paper was then reanalyzed by one of the authors (Fradin et al., 2017). The main conclusion of the authors is that the fast kinetics they observe are inconsistent with how quickly the Bicoid concentration appears to be interpreted in each nuclear cycle and that this may be due to the limitations of FCS. In the Fradin, 2017 paper it is hypothesized that this discrepancy may be resolved by considering that Bicoid molecules undergo “facilitated diffusion” in searching for their targets by sliding in 1D on DNA. However no new data has been generated to support this hypothesis. Thus these previous measurements do not provide any insight beyond speculation on the motilities with regard to the MSD analysis we have presented. In the revised manuscript we now cite these works in the context of discussing the complex nature of the TAMSD curves we measured and the underlying phenomenon that might explain them.

“Anomalous or sub-diffusive motion can result from a range of underlying physical interactions including aggregation, weak interactions with other proteins and chromatin, repetitive binding at proximal binding sites, among many other models (Woringer and Darzacq, 2018; Fradin, 2017)”.

On the other hand FCS measurements of His2B-EGFP (Battacharya et al., 2009) in the early *Drosophila* embryo have indeed revealed anomalous diffusion with the estimated coefficient of between 0.7 and 0.8, consistent with our estimates. We thank the reviewer for bringing these measurements to our attention and have now cited this paper in the main text as follows:

“Our estimation of the α value for His2B is in agreement with previous measurements using Fluorescence Correlation Spectroscopy (Bhattacharya et al., 2009).”

4) The fast dynamics of interphase hubs is a strong argument used by the authors in the overall message they want to deliver. This cannot be only based on qualitative arguments and the quantitative description of this dynamics has to be provided. In particular, the data presented Figure 5 and discussed in paragraph two of subsection “Zelda and Bcd form dynamic subnuclear hubs” have to be quantified.

We agree with the reviewer that a quantitative assessment of the dynamics of the hubs themselves is of high interest. We experimented with analytical methods to classify and track individual hubs to determine their size, movement, and persistence. However, commonly used metrics that rely on segmentation and tracking are too restrictive and require assumptions about the nature of these structures that we do not believe are valid. Developing novel metrics and analytical approaches to assess hub dynamics, treating them as the amorphous structures that they are, is an ongoing effort in our groups.

As an alternative to tracking and counting individual hubs, we have analyzed our movies of Zld and Bcd dynamics at sub-second frame rates by examining the frame-to-frame correlation of the overall nuclear distribution of these proteins. The results of this analysis match what we describe qualitatively — the spatial correlation between frames decays to background levels within ~3-7 seconds, consistent with typical hub lifetimes of a few seconds (Author response image 1). Interestingly, these correlations do not relax to the level of control experiments in which signals from different nuclei are compared. This is consistent with our observation that hub formation, while highly dynamic, is not random, and that subnuclear concentrations of TFs show unequal distributions when averaged over longer timescales (e.g., the preferential accumulation of Bcd protein at the *hb* locus).

**Author response image 1. respfig1:** Frame-to-frame correlation of nuclear protein fluorescence generated from movies with a 15 ms frame rate for Zelda and 210 ms frame rate for Bcd. Nuclei were manually segmented, fluorescent signal was normalized independently for each nucleus and each frame, and the pixel-wise correlation of nuclear signal between frames was calculated. Vectors of correlation coefficients were generated for series of frames beginning every by 420 ms, and coefficients at each time offset (+1 frame, +2, +3…) were averaged to produce the plots. The decay in correlation for the nuclei for each protein is shown in red; gray lines represent controls in which nuclei were permuted and correlations were measured between fluorescent signal in different nuclei. Shading represents standard error of the mean.

To further emphasize this point we have generated a new movie that shows more clearly the amorphous nature of Zelda hubs (Author response video 1):

**Author response—video 1. respvideo1:** 

5) Concerning the enrichment of Bcd and Zld at the hb transcription sites, the 1.3 versus 1.1 relative enrichment observed for Bcd seems to be due to lower background (Figure 7C). It would be important to comment on this and provide the data before normalization in the form of line profiles (one radius for X axis and intensity for the Y axis). This would help understand the data, get an idea of the noise in the measurements and determine whether the 1.1 to 1.3 increase could be meaningful.

It is unclear how a lower background would lead to a false measurement of enrichment of Bcd at the Hb locus compared to random sites in the nucleus — if anything this makes the case for enrichment stronger. To address the reviewers concerns we have now plotted the unnormalized line profiles in Author response image 2:

**Author response image 2. respfig2:** 

Also, it is not clear why the enrichment of Bcd binding events in Zld hubs could not be due to an increase k_off_. This needs to be clearly explained and rationalized.

We apologize for a typo where we meant to say that the enrichment is not due to an increase in residence time, meaning a decrease in koff, not an increase. This may account for some of the confusion. We have fixed this error in the revised manuscript.

As we stated in the main text, the increase in binding times is “is less pronounced than the overall enrichment of all Bcd binding events” suggesting that the enrichment we observe is due to an increase in kon. We have modified this sentence to further clarify our rationale as:

“We note that while there is an increase in long Bcd binding events in Zld hubs this effect is not large enough to account for the overall enrichment of all Bcd binding events, suggesting that Zld increases the time averaged Bcd occupancy at DNA binding sites primarily by increasing its local concentration (increasing kon) and not by increasing its residence times at its target sites (decreasing koff).”

Minor Comments:1) Introduction paragraph nine: beginning of the sentence is unclear given the cited reference.

We have now clarified this point by elaborating on why the cited reference is relevant as follows:

“Prompted by previous observations (Hannon, Blythe, and Wieschaus, 2017) that Bcd can bind to inaccessible chromatin on its own at high concentrations in the anterior but requires Zld to do so at low concentrations in the posterior, we examined the distribution of Bcd binding in nuclei lacking Zld and found that Bcd hubs no longer form (Mir et al., 2017)”

2) Figure 1B: yellow arrows are very difficult to distinguish from white arrows.

We have changed the color of the yellow arrows to green to help distinguish them from the white arrows.

3) Subsection “Zelda hubs are not stably associated with sites of active transcription” first paragraph: be more explicit on the number of sites given also the fact that Figure 7—figure supplement 1) does not at all provide any information related.

Locating transcription factor binding sites within metazoan regulatory sequences is a famously hard problem, and cannot be reliably achieved through sequence-based informatic searches alone. Several groups have used varied approaches to attempt to measure the cooperativity of the *hb* response to Bicoid concentration and infer the number of binding sites from resulting hill coefficients. For example, Xu et al., 2015, estimated a hill coefficient of ~6, while Lopes et al. (2012 *Developmental Biology*) reported a hill coefficient that increased from ~4 to more than 10 over developmental time. While it is thus difficult to nail down a specific, discrete number of binding sites, a consensus view is that 1) Bcd directly regulates *hb* transcription and 2) it does so through multiple DNA binding sites within *hb* regulatory regions. Therefore, we think that the ambiguous “multiple clustered Bcd binding sites” is an appropriate description.

4) Discussion section: transient binding (if this is really what is detected in this study) is not inconsistent with ChIP experiments showing increase occupancy of Bcd in Zld binding regions which are performed on fixed material and provide only a "cumulative" snap-shots (time scale of the fixation process) of the state of the bound regions. It might just be a matter of time-scale.

We agree with the reviewer that our results are consistent with published ChIP studies. ChIP is a measure of occupancy: the probability of a covalent cross-link forming between a locus and a protein species is proportional to the fraction of time that a site is bound by any protein of that type during the fixation period, not the residence time of individual molecules. Therefore, our model that rapid binding and unbinding of individual molecules results in high levels of time-averaged occupancy at target sites is entirely consistent with the strong and specific ChIP signal observed for these factors.

5) Discussion, paragraph five: not clear to what "previous reports" the authors refers to. Provide references.

We have added the appropriate references.

6) Discussion, paragraph five: better explain what is meant by "time-average accumulation" (and also the reference to the work on Bcd at hb).

We now have elaborated on this point in the revised manuscript:

“We suggest that what previous reports (Dufourt et al., 2018) have described as stable clusters are actually time-averaged accumulations (such as what we observe for Bcd at hb) resulting from insufficient temporal resolution or use of immunofluorescence data, rather than discrete sub-nuclear bodies. “

7) Discussion section: there is also a possibility that the hubs reflect some kind of structure formed when the transcription factors are inactive (the authors mention that they are "more pronounced at times when no transcription happens") and that the real interaction events responsible for activity are not detected because they are too rare. For instance: if the activation event requires 6 Bcd proteins bound at a given site, what is the probability that this event would be detected with photoactivable approach unless the interaction is long enough (what is the time-scale that would allow to capture these events).

It is possible that the functional nature of the hubs varies through the cell cycle as we hypothesize in the discussion. It is certainly possible that the hubs are playing a role in regulating chromatin accessibility or enhancer licensing when they are most pronounced at the beginning of the cell cycle and there is no transcriptional activity. However in this manuscript we are specifically examining the interaction of hubs with sites of active transcription. The reviewer is correct in their observation that if we were using a photoactivatable approach to try to study the enrichment of Bicoid at a target locus it would be unlikely for us to observe a specific binding event in a single experiment, for this reason we relied on bulk imaging to study the interactions between hubs and active loci.

8) Materials and methods, paragraph four: indicate the promoter allowing expression of H2B.

We have included this information in the revised manuscript:

“His2B-mEos3.2 was introduced as a supplemental transgene under the control of a ubiquitous pΔTubHA4C promoter (Zhang et al., 2013) and SV40 3’ UTR via PhiC31-mediated recombinase (Groth et al., 2004) into landing site VK33 (Venken et al., 2009).”